# LCDB 1.1: A Database Illustrating Learning Curves Are More Ill-Behaved Than Previously Thought

**Cheng Yan**[1]   **Felix Mohr**[2]   **Tom Viering**[1]

[1]Delft University of Technology   [2]Universidad de La Sabana

{c.yan-1, t.j.viering}@tudelft.nl   felix.mohr@unisabana.edu.co

## Abstract

Sample-wise learning curves plot performance versus training set size. They are useful for studying scaling laws and speeding up hyperparameter tuning and model selection. Learning curves are often assumed to be well-behaved: monotone (i.e. improving with more data) and convex. By constructing the Learning Curves Database 1.1 (LCDB 1.1), a large-scale database with high-resolution learning curves including more modern learners (CatBoost, TabNet, RealMLP, and TabPFN), we show that learning curves are less often well-behaved than previously thought. Using statistically rigorous methods, we observe significant ill-behavior in approximately 15% of the learning curves, almost twice as much as in previous estimates. We also identify which learners are to blame and show that specific learners are more ill-behaved than others. Additionally, we demonstrate that different feature scalings rarely resolve ill-behavior. We evaluate the impact of ill-behavior on downstream tasks, such as learning curve fitting and model selection, and find it poses significant challenges, underscoring the relevance and potential of LCDB 1.1 as a challenging benchmark for future research.

## 1 Introduction

In machine learning, a learning curve can refer to two types of curves. The *epoch-wise learning curve* (also known as *training curve*), depicts model performance versus training iterations or epochs. The *sample-wise learning curve* focuses on performance versus the amount of training data used for training [1]. Sample-wise curves provide a richer evaluation at multiple training data sizes [2]. These curves are useful for speeding up model selection and hyperparameter tuning using multi-fidelity techniques [3]. The curves are also useful to estimate how much data is needed to reach a particular performance [4, 5], providing insights into so-called scaling laws [6–9]. In this work, we focus exclusively on sample-wise learning curves and use the term *learning curve* to refer to them.

To effectively use learning curves, it is important to know their shape. When a suitable parametric formula can be assumed, it becomes possible to extrapolate the final performance from partial training data, thereby accelerating model selection. However, learning curve modeling remains challenging: existing parametric models often fail to outperform the simple strategy of selecting the best algorithm based on the last observed curve value [10, 11]. Furthermore, much remains unknown about the learning curve shape. Often, it is assumed that more data leads to better generalization performance [1, 10]. Such learning curves are called monotone: the loss decreases monotonically with more data. Similarly, learning curves are often assumed to be convex, meaning that there is an effect of diminishing returns: the more data we have, the less performance is improved by additional data. If a curve is monotone and convex, we call it well-behaved [1, 12, 13].

Learning curves can exhibit a variety of ill-behaved shapes in toy-settings, violating either monotonicity or convexity [1, 14]. Mohr et al. [10] studied whether such behaviors occur in non-toy settings. They collected the largest-scale database of learning curves, called the Learning Curves Database 1.0

Table 1: Ill-behaved learning curves in the wild on OpenML CC-18 classification datasets. The y-axis indicates the error rate on the validation set, and the x-axis represents the size of the training set. The line is the mean and the shaded area indicates one standard error; this applies to all subsequent learning curve plots. Peaking: error rate has a local maximum. Dipping: error rate worsens and does not recover. Phase transition: sudden improvement.

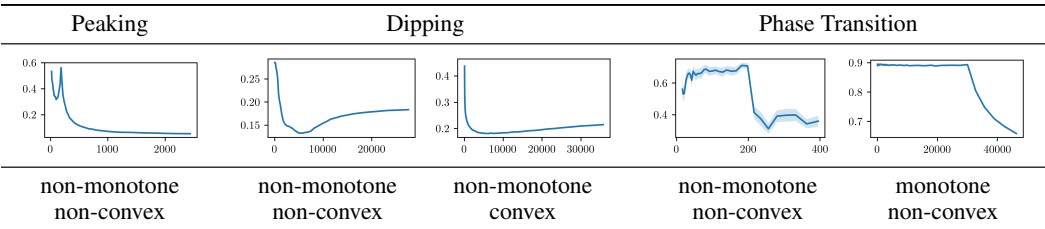

| Peaking | Dipping | | Phase Transition | |
|---------|---------|---|------------------|---|
| non-monotone non-convex | non-monotone non-convex | non-monotone convex | non-monotone non-convex | monotone non-convex |

(LCDB 1.0), from various learners, evaluated across hundreds of classification datasets [15, 16]. In LCDB 1.0, Mohr et al. [10] conclude: "We found that the large majority of learning curves is, largely, well-behaved, in that they are monotone, convex, and do not show peaking." We believe, however, that this conclusion is premature. Their analysis only shows that more extreme ill-behaviors are less frequent; however, it does not estimate how many are significant as a fraction of all curves.

The LCDB 1.0 also suffers from technical issues. It lacks resolution, which can make it difficult to find ill-behavior reliably. In LCDB 1.0, features were also not scaled. Feature scaling is a well-established and standard practice in machine learning that improves training stability and model performance [17–20]. Therefore one may also wonder if the ill-behavior may disappear simply by scaling. Indeed, we find that sometimes feature scaling makes a curve well-behaved, see Figure 1a. Besides, we find that LCDB 1.0 suffers from missing data and a minor data-leakage issue. These issues illustrate the need for a new database and deeper analysis of the prevalence of ill-behaviors.

We introduce the Learning Curves Database 1.1 (LCDB 1.1) which addresses the aforementioned limitations. We incorporate four-times more training set sizes, see Figure 1b. This increases the resolution of the curves, which allows us to find more ill-behavior. We argue that, depending on how learning curves are used, data-leakage is sometimes acceptable. Therefore, we introduce two database versions, one with data-leakage and the other without, and also incorporate different feature scalings. In case a performance value is missing due to an error, we justify and document it. Besides, we include the OpenML CC-18 datasets [21], which are more carefully curated datasets, and some more modern tabular data learners, including boosting (CatBoost [22]), deep learning (TabNet [23], RealMLP [24]), and foundation models (TabPFN v2 [25, 26]).

Next to providing a new database, we provide a richer analysis of the ill-behaved learning curves. We develop methods to detect whether a learning curve is significantly non-monotone or non-convex and also measure the size of violations. Besides, we also identify other learning phenomena, such as peaking, dipping, and phase transitions, see Table 1. We demonstrate that these ill-behavior are significant and happen often for particular learners. Feature scaling cannot mitigate these ill-behaviors in most of the cases, ruling out that these issues can be easily resolved.

So, we show that these ill-behaviors are significant, but are they also relevant for downstream tasks such as learning curve fitting and model selection? We conduct learning curve fitting experiments using parametric formulas. We investigate the relation between curve fitting and ill-behavior. Most

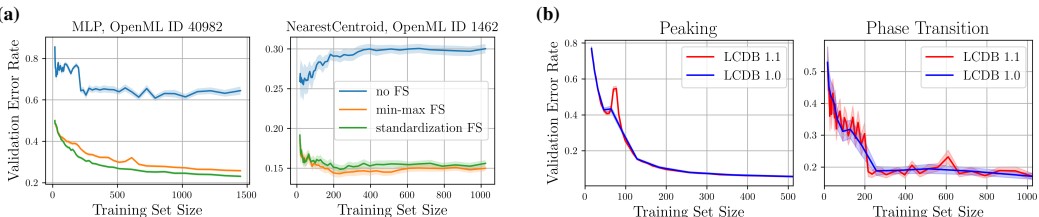

Figure 1: Motivation for new LCDB 1.1 features. (a) Feature scaling can mitigate an ill-behaved learning curve. (b) Low-resolution curves may omit certain phenomena or render them less apparent.

parametric curve models lack the flexibility to model non-monotone and non-convex behavior [27], and indeed we confirm that parametric modeling is significantly more difficult on ill-behaving curves. Learning curves can be used for multi-fidelity model selection using Successive Halving (SH). We find that the crossing curves in our database also make multi-fidelity model selection more challenging. Thus, we illustrate the relevance of these ill-behaviors for downstream tasks, and the unique challenges posed by sample-wise learning curves.

In summary, we create a new and improved database. We perform a more in-depth analysis regarding ill-behaved curves. These analyses illustrate what is inside our database, and therefore we call these database analyses (DA), that we pose as questions. DA1: How many learning curves are significantly ill-behaved and which learners are responsible? DA2: How does feature scaling affect ill-behavior? DA3: How do ill-behaved curves affect learning curve extrapolation? DA4: How does learning curve crossing affect model selection using successive halving?

We discuss the design of LCDB 1.1 in Section 3. Section 4 describes how to robustly detect ill-behaviors. The experimental setup and the results are in Section 5, and we end with discussion and conclusion in Section 6. First, we cover preliminaries and related work.

## 2    Preliminaries and Related Work

**Sample-Wise Learning Curves: Theory and Practice.**    A sample-wise learning curve returns a performance $C(n)$ versus the training set sizes $n$ used. Here we discuss its theoretical definition and how it is computed in practice, focusing on classification tasks. Let $S_n$ be the training set, consisting of features $x \in \mathbb{R}^d$ and corresponding class labels $y$, thus $S_n = \{(x_1, y_1), \ldots, (x_n, y_n)\}$. We assume that there exists a distribution $P$ from which $(x, y)$ are independent and identically distributed samples. $A(S_n)$ is a learning algorithm trained on $S_n$. Let $R(A(S_n))$ be the risk, which indicates its loss in expectation on data from $P$. For classification commonly the zero-one loss is used, in this case, $R(A(S_n))$ is the error rate. The theoretical mean learning curve $C(n)$ is defined as:

$$C(n) = \mathbb{E}_{S_n \sim P} R(A(S_n)), \tag{1}$$

The curve is computed over a number of training set sizes, e.g. $n_1, n_2, \ldots$, where we call the training set size *anchor*. The risk is an expectation that relies on an integral over the true data distribution $P$, which is unknown. Therefore, we estimate the risk using performance on held out data (test data). The expectation over $S_n$ is approximated using multiple repeats with different train and test sets. By using multiple repeats $K$, we obtain multiple estimates of the risk, $\hat{R}_n^r$, where $n$ indicates the training set size and $r$ is the repetition. We estimate the mean learning curve as $\hat{C}(n) = \frac{1}{K} \sum_{r=1}^{K} \hat{R}_n^r$. One decides $n_1, n_2, \ldots, n_N$, typically based on the dataset size. $N$ is the amount of anchors in a curve.

**Ill-Behaved, Non-Monotone, and Non-Convex Learning Curves.**    Several synthetic learning problems illustrate that more data does not lead to better performance [14, 28], we find such examples in carefully curated CC-18 datasets [21] (see Table 1). Peaking is such a violation, where the error rate initially decreases as the training set size increases, then rises to a peak before decreasing again [29, 30], and is also called sample-wise double descent [5, 31]. Peaking has been proven to occur for the Fisher classifier, and the peak effect is most severe when the training set size is equal to the dimensionality $d$ [32, 33]. Double descent describes a similar phenomenon when plotting the error versus the capacity of the model [1, 34]. Other cases of monotonicity and convexity violations include dipping and phase transitions. Loog and Duin [35] introduce the concept of dipping. The error rate initially improves, then increases without recovering, even in the limit of infinite amounts of data. Dipping has been observed for decision trees in error rate learning curves [36, 37], and for KNN in AUC learning curves [38]. A phase transition means that model performance improves abruptly, causing a distinct drop in the learning curve. Phase transitions in machine learning were studied mostly in theory [1, 39], and we are not aware of any examples on real-world datasets before this work. Beyond classification, many of the observed learning curve irregularities, such as non-monotonicity, may also arise in regression problems [14, 40, 41] and even unsupervised learning [42], challenging the assumption that more data always helps.

**The Learning Curves Database 1.0 (LCDB 1.0).**    The LCDB 1.0 [10] includes classification learning curves of various learners on numerous datasets from the OpenML platform [43–45]. Some

Table 2: Main innovations of LCDB 1.1 compared to LCDB 1.0.

| Database | Preprocessor | Feature Scaling | Anchor Resolution | #Learners | #Datasets | Missing | Missing Reason |
|---|---|---|---|---|---|---|---|
| LCDB 1.0 | with data-leakage (dl) | none | $\lceil 16 \cdot 2^{k/2} \rceil$ | 20 | 196 (claim 246) | 12% (30%) | unknown |
| LCDB 1.1 | with and without dl | none, min-max, standard | 4 times denser | 32 | 265 | 4% | documented |

curves in LCDB 1.0 are missing, resulting in actually fewer than 196 datasets, possibly due to incompatibilities with sparse matrices and long compute times. Data leakage occurs because the feature imputer was fitted on the complete data. We resolve these issues with the LCDB 1.1.

**Other Datasets and Relation to Deep Learning.** Task-Set [46], LCBench [47], and BUTTER [48] are datasets containing epoch-wise curves of neural networks. They are not comparable to ours, since we study sample-wise learning curves. In deep learning, the scaling law literature focuses on much larger training sets and presents much sparser learning curves [7–9]. LCDB 1.1 instead focuses on tabular data, where many classical algorithms remain competitive with deep learning [49–52], and where datasets are typically smaller. Furthermore, tabular data offer unique challenges: these datasets are rare [53], and columns are often incomparable across datasets, complicating knowledge transfer [54, 55]. Meanwhile, tabular data are crucially important for industry [56]. Insights into learning curves can help estimate how much tabular data is needed [5] which is important when data is costly.

**Learning Curve Fitting.** Learning curves are usually modeled by parametric formulas [1, 8, 57]. Popular functions are exponential and power laws, which are motivated by the well-behaved assumption [1]. Learning curve fitting can be used to estimate the amount of data needed [5]. Mohr et al. [10] identified that parametric models with 4 parameters seem to perform best for interpolation, such as POW4, where $\hat{C}(n) = a - b(d + n)^{-c}$. The most widespread technique is least square curve-fitting using Levenberg-Marquadt [28], more advanced techniques use Bayesian techniques and neural networks [58–60]. We investigate the effect of ill-behavior on least square curve fitting.

**Multi-Fidelity Model Selection.** Successive Halving [61] (SH) is a method to speed up model selection. It uses a fidelity; the higher the fidelity, the more accurate model performance is estimated. The fidelity can represent the amount of epochs used or the amount of training data. SH is iterative, evaluating model performances first at low fidelities and moving to higher fidelities afterward. In each round, a percentage of the learners that perform poorest are dropped, and the fidelity is increased. SH can be combined with learning curve extrapolation for both learning curves [13] and training curves [58]. If learning curves often cross, SH may perform suboptimal, which we will investigate. Various multi-fidelity methods exist [58, 62–67], we use SH since it is popular and interpretable.

## 3 The Improved Learning Curves Database 1.1

Table 2 gives an overview of the main differences between LCDB 1.1 and 1.0. First, we discuss data splitting, preprocessing and we justify two the versions with and without data-leakage. We briefly discuss dataset and learner selection, and end with metrics, reproducibility, and some statistics. Some details equal to LCDB 1.0 are omitted (see Appendix A). The LCDB 1.1 is publicly available.[1]

**Data Splitting.** We use multiple train-validation-test sets to enable the simulation of model selection using nested cross validation. Selection can be done using validation and evaluation using the test set. We use 5 inner and 5 outer seeds to create these datasets. Let $D$ be the complete dataset, then

$$D \xrightarrow[\text{outer seed } m]{\text{outer split}} \left( D_{\text{train-val}}^{(m_o)}, D_{\text{test}}^{(m_o)} \right) \quad \text{then} \quad D_{\text{train-val}}^{(m)} \xrightarrow[n \text{ random seed}]{\text{inner split}} \left( D_{\text{train}}^{(m_o, m_i)}, D_{\text{val}}^{(m_o, m_i)} \right)$$

where the superscripts indicate outer ($m_o$) and inner ($m_i$) seeds. LCDB 1.0 uses training anchors $n_k = \lceil 16 \cdot 2^{k/2} \rceil$, where $k \in \{0, 1, 2, ...\}$. The LCDB 1.1 uses $n_k = \lceil 16 \cdot 2^{k/8} \rceil$ resulting in four times higher resolution. Further details are as in LCDB 1.0 (see Appendix A).

---

[1]LCDB 1.1 dataset: `https://doi.org/10.4121/3bd18108-fad0-4e4c-affd-4341fba99306`

Table 3: The 32 learners in LCDB 1.1 FULL (265 OpenML datasets, no scaling version), their estimated ill-behaved (non-monotone or non-convex) ratio, and their abbreviations.

| Learners (Abbreviation) | Ill-behaved | Learners (Abbreviation) | Ill-behaved |
|---|---|---|---|
| CatBoost [22] | 1.5% | Complement Naive Bayes (ComplementNB) | 8.3% |
| Decision Tree (DT) | 1.5% | Passive Aggressive (PA) | 9.4% |
| TabPFN v2 [26] * | 1.5% | Mix Complement Naive Bayes (MixComplementNB) | 10.2% |
| Extra Tree (ET) | 1.9% | Mix Multinomial Naive Bayes (MixMultinomialNB) | 10.6% |
| ensemble Gradient Boosting (ens. GB) | 1.9% | RBF Support Vector Machine (SVM_RBF) | 15.8% |
| ensemble Random Forest (ens. RF) | 3.0% | Ridge Regression Classifier (Ridge) | 17.0% |
| Stochastic Gradient Descent Classifier (SGD) | 3.4% | Mix Gaussian Naive Bayes (MixGaussianNB) | 21.5% |
| ensemble Extra Trees (ens. ET) | 3.4% | Gaussian Naive Bayes (GaussianNB) | 24.9% |
| Perceptron | 3.8% | Multilayer Perceptron (MLP) | 27.9% |
| K-Nearest Neighbors (KNN) | 3.8% | Bernoulli Naive Bayes (BernoulliNB) | 28.3% |
| RealMLP [24] ** | 5.3% | Mix Bernoulli Naive Bayes (Mix BernoulliNB) | 28.7% |
| Logistic Regression (LR) | 5.3% | Linear Discriminant Analysis (LDA) | 37.7% |
| Linear Support Vector Machine (SVM_Linear) | 5.7% | Quadratic Discriminant Analysis (QDA) | 45.7% |
| Polynomial Support Vector Machine (SVM_Poly) | 7.9% | Sigmoid Support Vector Machine (SVM_Sigmoid) | 58.1% |
| Multinomial Naive Bayes (MultinomialNB) | 7.9% | Dummy Classifier (Dummy) | 60% |
| Nearest Centroid (NC) | 7.9% | TabNet [23] | 74.3% |

* The reported results cover 210 out of 265 datasets with the maximum curves length less than 10k, due to the fact that TabPFN v2 only supports datasets with up to 10k training samples, 500 features, and 10 classes. Note that some datasets included in LCDB 1.1 were used in designing its prior.
** Some LCDB 1.1 datasets were used in the meta-train benchmark for designing and meta-tuning RealMLP.

**On Preprocessing and Data Leakage.** In LCDB 1.0, the imputer was fitted on the whole dataset. In LCDB 1.1, we apply: no scaling (abbreviated as "noFS"), min-max scaling, or standardization of features. Because of this additional preprocessing, it is even more important to discuss how to fit the preprocessor and data-leakage. When learning curves are applied for model selection and hyperparameter tuning, the goal is to reduce computation time. We can assume the user has access to the complete dataset. Fitting the preprocessor on the whole dataset can then lead to better performance and stability, and data-leakage is acceptable. However, when trying to estimate how much data is needed, we cannot assume the user has access to all data. Thus, in this case, data-leakage is not acceptable. We therefore construct two LCDB 1.1 variants, with and without data-leakage. To prevent data-leakage, preprocessors are fitted on the train set. We compare these versions in Appendix B.

**Dataset and Learner Selection.** In LCDB 1.0, we observe that some datasets are overly easy, resulting in flat learning curves that are already converged at the first anchor. Therefore, we include all datasets of the OpenML-CC18 benchmark [21] in LCDB 1.1, called LCDB 1.1 CC-18. This benchmark was carefully curated, filtering out datasets that are overly easy, amongst other issues. The complete LCDB 1.1, referred to as LCDB 1.1 FULL, combines CC-18 with all datasets of LCDB 1.0.

The LCDB 1.1 has 32 learners, see Table 3. The dummy predicts the majority class and provides a weak baseline. One-hot features violate assumptions of Naive Bayes [68], to that end we introduce mixed Naive Bayes learners. Moreover, we incorporate a broader set of modern tabular learners: the boosting model CatBoost [22], deep learning models such as TabNet [23] and RealMLP [24], and the foundation model TabPFN v2 [26]. According to Erickson et al. [69], CatBoost remains a strong state-of-the-art model by default, while RealMLP achieves state-of-the-art performance after tuning and ensembling. TabNet is a popular deep learning baseline [23], and TabPFN v2 is a well-performing foundation model for tabular data [26]. See Appendix A for all added learners.

All these modern learners claim robustness to differently scaled features; therefore, we only include their no scaling (noFS) and no data-leakage variants in LCDB 1.1 FULL (except TabPFN, which does not explicitly address feature scaling) [22–24, 26]. Regarding categorical features, RealMLP and TabPFN use one-hot encoding, whereas for CatBoost and TabNet we follow their suggested practice of directly feeding categorical features into the model. For implementation details see Appendix A.

**Metrics, Reproducibility, and Database statistics.** We use Python, scikit-learn [70], a docker image, save all package versions and provide all the code for reproducibility.[2] We fix the seed of the learner to make them reproducible. We compute: error rate, F1, AUC, and log-loss for the validation and test sets, and we store learners' scores or probabilistic outputs. When training fails, we record the error message and set the performance to Not-A-Number (NaN). Table 4 shows the proportions of different curve shapes (their detections are discussed in the next section). Missingness refers to NaN

---

[2]https://github.com/learning-curve-research/LCDB-1.1

values and is mostly caused by Naive Bayes learners that cannot handle negative features. While not the main point of this work, one may wonder which learners perform best; see Appendix C.

## 4 Robustly Measuring Monotonicity, Convexity, Peaking and Dipping

In this section, we introduce the methods to detect and also measure monotonicity violations, convexity violations, peaking and dipping curves. However, we first criticize the methods of Mohr et al. [10]. They analyze monotonicity and convexity, but only consecutive anchors are compared. This will miss violations that happen over longer ranges, which is why we compare all anchors. We also check for significance, and since neighboring training set sizes may not yield significant differences, comparing all pairs is even more crucial. The convexity measure of Mohr et al. [10] treats the anchors as linearly spaced, ignoring that they are defined in logarithmic scale, which leads to incorrect conclusions. Our method incorporates the anchor scale. We use a hypothesis test to ensure detections are significant, where we are pessimistic, e.g. we only find violations if we are confident, otherwise we assume the curve is well-behaved. This aligns with the prior belief that most curves are well-behaved following literature [1, 10]. We assume a metric $C$ where lower means better.

A monotonicity violation means that the curve does not always improve with more data, see Figure 2.

**Definition 1** (**Monotonicity Violation Error**). *The largest increase between any anchor pair is*

$$\epsilon_{mono} = \max\left(0, \max_{1 \leq i < j \leq N} \left(C(n_j) - C(n_i)\right)\right). \tag{2}$$

The violation error $\epsilon_{\text{mono}}$ measures the largest size of the violation and is zero if there is none.

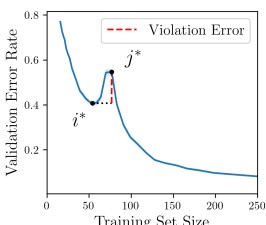

To detect violations from empirical learning curves, we use the following procedure. We compute $\hat{\epsilon}_{mono}$ using the empirical curve means and find the pair $(i^*, j^*)$ that maximizes Equation 2. If $\hat{\epsilon}_{mono}$ is zero, we classify the curve as monotone. If $\hat{\epsilon}_{mono} > 0$ we check the significance of the violation. We compare the empirical distributions $\hat{R}^r_{n_{i^*}}$ and $\hat{R}^r_{n_{j^*}}$ using a paired one-sided $t$-test with Bonferroni correction. Paired, because the same inner and outer seeds are used, and one-sided because we only care about violations in one direction. The Bonferroni correction corrects for multiple testing, assuring we do not find too many violations due to noise. This correction is necessary because identifying the maximum among anchor pairs implicitly involves multiple comparisons. We correct on a curve-level for all anchor pairs. The corrected significance level is $\alpha' = \frac{\alpha}{N(N-1)/2}$, where $\alpha$ is the original significance level. If the $p$-value is smaller than $\alpha'$ we classify the curve as non-monotone.

Figure 2: Monotonicity Violation

A function is convex if its linear interpolation is always above the function itself. If the curve is above its linear interpolation, this is a convexity violation, see Figure 3.

**Definition 2** (**Convexity Violation Error**). *The linear interpolation of a curve from anchor $n_h$ to $n_j$ evaluated at $n_i$ is:* $C_{interpolated}(n_i; n_h, n_j) = \frac{n_j - n_i}{n_j - n_h} C(n_h) + \frac{n_i - n_h}{n_j - n_h} C(n_j)$. *We define*

$$\epsilon_{conv} = \max\left(0, \max_{1 \leq h < i < j \leq N} \left(C(n_i) - C_{interpolated}(n_i; n_h, n_j)\right)\right). \tag{3}$$

The violation error $\epsilon_{conv}$ measures the largest convexity violation and is zero if there is none.

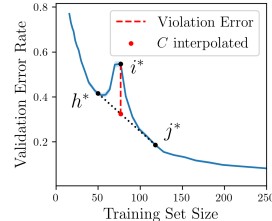

We detect a convexity violation from empirical data using the following procedure. First, we evaluate $\hat{\epsilon}_{conv}$ using the empirical means of the learning curve. If $\hat{\epsilon}_{conv} < 0$, the curve is classified as convex. If $\hat{\epsilon}_{conv} > 0$, we check the significance of the violation. First, we find the maximizers $(i^*, j^*, h^*)$ of Equation 3. For each repeat, we interpolate the curve linearly, to obtain the empirical distribution of the interpolated curve. The interpolated and actual distributions are compared using one-sided paired $t$-test. We correct for the triplet comparison using Bonferroni; thus if the $p$-value is smaller than $\alpha' = \frac{\alpha}{N(N-1)(N-2)/(3!)}$, we classify the curve as non-convex.

Figure 3: Convexity Violation

**Definition 3** (**Peaking Phenomenon**). *Peaking occurs if there exists a triplet of indices,* $1 \leq h < i < j \leq N$*, such that:*

$$C(n_i) > C(n_h) \quad and \quad C(n_i) > C(n_j). \tag{4}$$

In this case, $C(n_i)$ forms a local peak, indicating that the model's performance temporarily degrades and subsequently recovers as more data is added.

**Definition 4** (**Dipping Phenomenon**). *Dipping occurs if there exists an index* $i$*,* $1 \leq i < N$*, such that:*

$$C(n_i) < C(n_N). \tag{5}$$

$N$ denotes the amount of anchors in a curve. This indicates a sustained degradation of model performance, with no recovery observed as more data is added.

Lastly, we describe how peaking and dipping are detected. Peaking is characterized by a combination of convexity and monotonicity violations: we first locate a convexity violation at $(h^*, i^*, j^*)$, and then verify a monotonicity violation between $h^*$ and $i^*$ and we check for significant improvement between $i^*$ and $j^*$ (similar to violation error detection, but instead checking for improvement). If all 3 conditions are satisfied, the curve is classified as peaking. Dipping corresponds to a monotonicity violation with $j$ fixed as the last anchor $N$.

## 5 Results

Here we discuss the database analyses (DA) that we perform and the experimental setup.

### 5.1 Experimental Setup

Both QDA and the Dummy classifier are excluded due to reproducibility issues and the lack of meaningful learning behavior, respectively. We do not conduct analyses of mixed Naive Bayes methods, as their curves are largely indistinguishable from standard Naive Bayes (Appendix C). We always focus on error rate learning curves in LCDB 1.1 CC-18, since its selection of datasets is more carefully curated. Results on LCDB 1.1 FULL are similar (see Appendix D). **DA1 and DA2.** A significance level of $\alpha = 0.05$ is used throughout, and the curves are estimated using the validation set. Since we have 5 inner and 5 outer seeds, the learning curves are estimated from 25 repeats, which are aggregated together. To compare with LCDB 1.0, we interpolate the LCDB 1.0 curves to have the same length to ensure Bonferroni corrections are comparable. **DA3.** We closely follow the curve fitting methodology of LCDB 1.0 [10] and also use the validation set. We use the parametric models POW4, MMF4, and WBL4 since they performed best. Flat curves are filtered because they are overly easy to fit, leading to very small MSEs. To detect them, we scale all learners' curves to [0,1] range and classify it as flat if the maximum minus minimum value is below 0.05. **DA4.** We run successive halving to perform model selection, where the fidelity is determined by the anchor. Model selection is done using the validation set, and the selected model is evaluated using the test set.

### 5.2 DA1: How Many Curves Are Significantly Ill-Behaved and Which Learners Are Responsible?

An overall picture of the violations can be observed in Table 3 and 4. In this section, we only discuss the no feature scaling case ("no FS"). A substantial amount of curves is non-monotone (9.9%) and non-convex (11.5%), leading to 14.9% ill-behaved curves. This is significantly larger than the significance level $\alpha$, ruling out that these curves ill-behaviors are purely caused by noise. Note that the LCDB 1.0 barely passes this bar, underlining the need for a higher resolution database like LCDB 1.1 to detect all ill-behaviors. Peaking is responsible for 5.0% and dipping is responsible for 6.1%. The amount of flat curves is reduced for the CC-18 version compared to the FULL version as expected due to more careful curation.

In Figure 4, we visualize the ill-behaviors per learner. Learners that have less than 5% of any of the ill-behaviors are omitted, for a full overview see Appendix D. Again, we discuss the case of no feature scaling. The MLP can exhibit surprising learning curve shapes that we classify as phase transitions (see examples in Appendix E.1). Additionally, we observe several peaking caused by artifacts arising from the interplay between batch size and training set size (see also Appendix E.1). The Sigmoid

Table 4: Ill-behavior statistics of the LCDB 1.1 variants and LCDB 1.0. Since we use a significance level of 5% to detect ill-behaviors, we can expect 5% false positives (in the worst-case). Therefore, only numbers larger than 5% are significant, which the LCDB 1.0 barely satisfies. Note: "no FS" results include the statistics of 4 more modern learners.

| Shapes / Database | LCDB 1.1 CC-18 (72) | | | LCDB 1.1 FULL (265) | | | LCDB 1.0 (196) |
|---|---|---|---|---|---|---|---|
| | no FS | min-max FS | standardization FS | no FS | min-max FS | standardization FS | no FS with interp. |
| Missing | 2.1% | 0.0% | 8.7% | 3.0% | 0.0% | 8.7% | 11.9% |
| Flat | 7.1% | 5.8% | 3.4% | 9.9% | 7.9% | 5.3% | 5.2% |
| Non-Monotone ($\neg$ M) | 9.9% | 11.2% | 9.2% | 9.6% | 11.1% | 9.5% | 5.1% |
| Non-Convex ($\neg$ C) | 11.5% | 9.4% | 8.4% | 12.3% | 10.0% | 8.8% | 5.7% |
| Ill-behaved ($\neg$ M $\cup$ $\neg$ C) | 14.9% | 13.5% | 11.2% | 15.4% | 14.3% | 11.8% | 8.1% |
| Peaking | 5.0% | 3.3% | 2.9% | 5.7% | 3.7% | 3.2% | 2.5% |
| Dipping | 6.1% | 8.5% | 6.3% | 6.9% | 9.6% | 7.2% | 4.6% |

SVM is a notably ill-behaved learner, showing many monotonicity and convexity violations, of which most can be classified as dipping. The RBF SVM is more well-behaved but does show some peaking. Note that the statistical stringency differs across ill-behaviors. For example, Sigmoid SVM shows more dipping than monotonicity violations, we return to this issue in Section 6.

We also observe peaking for LDA and the Ridge classifier. This can be expected because Ridge and LDA are closely related to Fisher [71, 4.3] [72, 4.1.5] which is known to peak. Surprisingly, KNN, Naive Bayes, and Nearest Centroid also do not always behave well. For Nearest Centroid, it was known it could dip [35], but this was never observed outside of toy settings. It can be concluded that a few learners are in fact responsible for the most ill-behaving curves. The most well-behaved learners are tree-like and ensemble learners (see also Table 3).

While many modern learners, such as CatBoost, RealMLP, and TabPFN v2, tend to exhibit well-behaved learning curves, this is not always the case. The notably ill-behaved learner TabNet exhibits substantial non-convexity in its learning curves (see Appendix D), which mostly appears to stem from phase transition phenomena (see Appendix E.2). We suspect that TabNet was designed for large datasets, and that its default hyperparameters are not suitable for small dataset sizes. Indeed, we find that generally for large training set sizes, TabNet performs well (see Appendix C).

### 5.3 DA2: Can Feature Scaling Mitigate Ill-Behavior?

To understand the impact of feature scaling, we now compare the results across scaling techniques. Table 4 indicates that feature scaling marginally reduces the amount of ill-behavior. From Figure 4

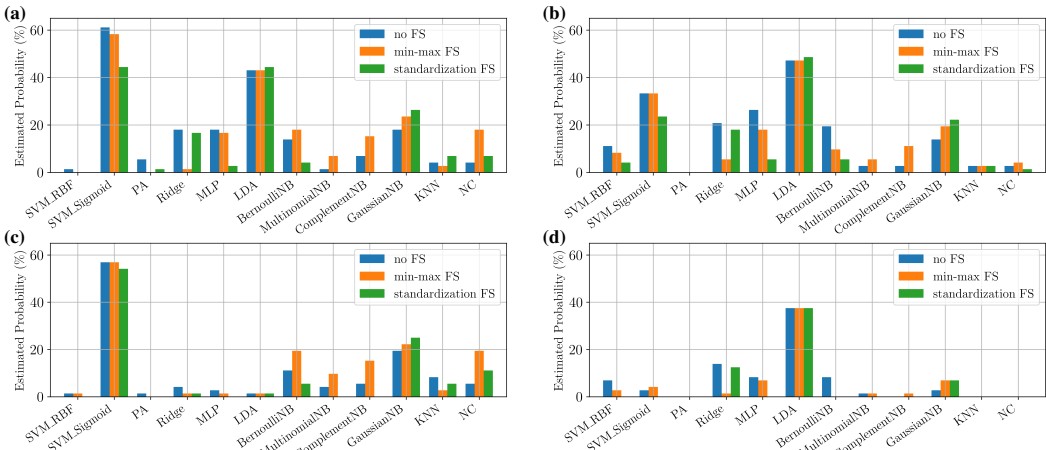

Figure 4: Estimated probability (%) of different ill-behaviors, (a) Monotonicity Violation, (b) Convexity Violation, (c) Dipping, (d) Peaking, for learners with different feature scalings. For all results see Appendix D. Observe that feature scaling for most learners does not lead to significant changes. Ridge and MLP improve significantly, while NC becomes more ill-behaved.

we observe that for most learners, feature scaling does not resolve ill-behavior. The Sigmoid SVM becomes slightly more monotone and has fewer peaks, but is still significantly ill-behaved. While preprocessing does not reduce ill-behavior, the SVM absolute performance improves notably and training becomes more stable after scaling is applied (see Appendix E.3 for details). Nearest Centroid and some Naive Bayes models are the only models that become significantly more ill-behaved with feature scaling. Note that GaussianNB is not entirely invariant to feature scaling, due to the way it calculates the variance for numerical stability. The biggest reductions in ill-behavior occur for the Ridge classifier and MLP. Ridge becomes almost completely monotone and without peaks when using min-max scaling, but not with standard scaling. The MLP improves significantly when using standard scaling, largely resolving ill-behaviors, however, min-max scaling does not always help the MLP. We confirm LDA is insensitive to feature scaling, and find the peak occurs when the training set size is approximately equal to the dimensionality (Appendix E.4) in line with peaking literature.

A further analysis showing which datasets are responsible for ill-behavior can be found in Appendix E.5. The ill-behavior seems to occur on almost all datasets, and in particular, it is not possible to attribute ill-behavior to a small number of datasets. In conclusion, few models become more well-behaved with preprocessing, and the type of preprocessing that helps can differ per model.

## 5.4 DA3: How Do Monotonicity and Convexity Violations Affect Curve Fitting?

In Figure 5a we show how the curve fitting performance is affected by convexity and monotonicity violations. We focus here on the results for the parametric formula POW4 (power law). For MMF4 and WBL4, results are similar, see Appendix F. Performance is measured using the mean squared error (MSE) on the fitted points (interpolation). The mean of the log MSE for monotone curves is over ten times smaller than for non-monotone curves, and the same applies to convex versus non-convex curves. Figure 5b visualizes the MSE versus the violation error. The results reveal a clear positive correlation between violation error and MSE. Our findings clearly show that parametric model fitting is significantly harder for non-monotone and non-convex curves, establishing LCDB 1.1 as a challenging benchmark for evaluating learning curve modeling methods.

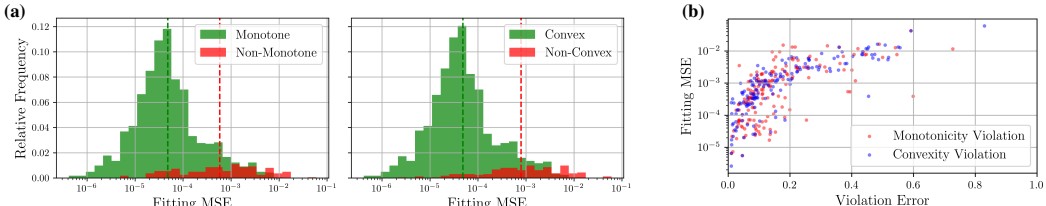

Figure 5: Ill-behaved learning curves pose challenges for curve fitting. (a) The distribution of fitting MSE when applying a parametric model to monotone vs. non-monotone (left) and convex vs. non-convex (right) learning curves. The dashed lines represent mean of the log MSE. Ill-behavior leads to to significantly larger MSE. (b) Larger violation sizes (x-axis) coincide with larger MSE (y-axis).

## 5.5 DA4: How Do Crossing Learning Curves Affect Model Selection?

Here, we choose two sets of learners and run Successive Halving (SH) on them to perform model selection with the training set size as fidelity, to investigate the influence of crossing curves. We determine 5-subsets of learners, one set of learners that often cross, and one set of learners who rarely cross; see Figure 6a. On both sets of learners we run SH, the results are shown in Figure 6b. In the left figure, we show how often the best algorithm is found. However, since the final performance differences of learners may be very similar, we complement this figure with the regrets on the right. Regret is the final error rate of the chosen learner minus the minimum of the final error rate over the learners in the subset (note the log-scale). Results for more settings are given in Appendix G. In the group of learners whose curves rarely cross (blue), the algorithm almost always picks the best or at least runner-up. For learners that frequently cross this is less often the case. The regrets show a similar pattern. As such, we can observe that crossing curves make model selection using SH significantly more challenging, highlighting the usefulness of LCDB 1.1 as a challenging benchmark for evaluating multi-fidelity model selection strategies.

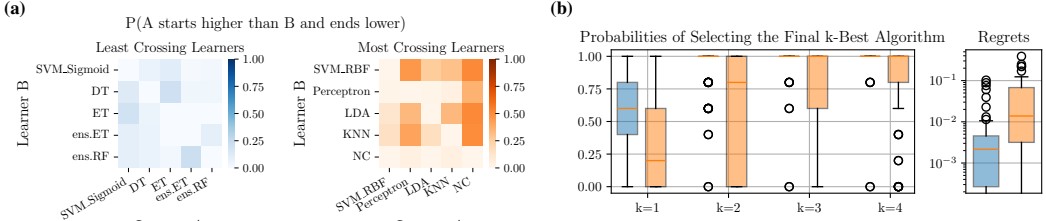

Figure 6: (a) Learning curve crossing probabilities of the two learner subsets (blue and orange), heatmap indicates probability. (b) Results of Successive Halving (SH) applied to blue and orange learner subsets. Fewer crossings lead to better SH performance. More results are in Appendix G.

## 6  Discussion

In contrast to Mohr et al. [10], we do find significant amounts of ill-behavior using our improved LCDB 1.1. While peaking and dipping were previously known for LDA, Ridge, Nearest Centroid, and MLP [1], their occurrence in realistic settings was not established. For the Sigmoid SVM, Naive Bayes, and KNN, it was not known ill-behavior was possible (either in toy or realistic settings). Ensemble methods have very well-behaved curves, yet we observe severe dipping on OpenML dataset 41027 [73]. The causes of these ill-behaviors remain unclear and present a challenging open problem.

The shape analysis is challenging. Note that ill-behavior may change if curves are longer (especially dipping). It is also difficult to maintain statistical rigor and consistency; this is because the Bonferroni correction imposes different levels of stringency, for instance, when testing monotonicity (two anchors) and convexity (three anchors). Bonferroni is also quite pessimistic, and some subjective choices had to be made. For example, peaking can also be detected differently (see Appendix H), yet results are similar. Moreover, we have performed additional analysis using slightly less conservative method called Holm's Step-Down Procedure (Holm's method) [18], in the sense that it will reject more null hypotheses, typically resulting in fewer Type II errors. This slightly increases the proportion of ill-behaved cases to 19%, but preserves overall consistency (see Appendix I). An analysis using E-values [74] or controlling the false discovery rate [75] may alleviate inconsistency issues, but such an analysis is non-trivial, going beyond our main point.

We have tried our best to make LCDB 1.1 fully reproducible, by using a docker container and fixed python package versions, yet we find that one LDA curve and several QDA curves are non-reproducible, likely due to numerical non-determinism of the singular value decomposition. For this reason, we exclude the QDA learner from the analysis but include its curves in the database.

The next step is to investigate whether ill-behavior persists under hyperparameter tuning, which we leave for future work, since collecting learning curve data with tuning is computationally expensive (LCDB 1.1 already required 800K CPU hours; see Appendix J). This investigation is particularly relevant for models that may exhibit strong sensitivity to hyperparameter settings, such as TabNet. Moreover, hyperparameters in Scikit have reasonable defaults determined by the community, and as such the ill-behavior observed remain surprising and relevant, especially when persistent to different scalings. Although our empirical analysis is scoped to error rate learning curves for classification, the observed ill-behaviors are not confined to this setting. It is therefore valuable to examine alternative evaluation metrics, such as AUC, F1 score, and log-loss, all of which we provide in LCDB 1.1.

## 7  Conclusion

In conclusion, we introduce the Learning Curves Database 1.1 (LCDB 1.1). This database is more reproducible, of higher resolution, and has multiple types of preprocessing (with and without data-leakage), as well as more modern learners such as CatBoost, TabNet, RealMLP, and TabPFN, making it a valuable database for the community to study learning curves. Moreover, we carefully study ill-behavior and find that a significant amount, 15%, of the curves exhibit ill-behavior while some learners misbehave more frequently than others. Feature scaling rarely solves this problem and in some cases can make it worse. Lastly, we demonstrate the impact of ill-behavior on downstream tasks, underscoring the practical implications. We hope that LCDB 1.1 facilitates new investigations of ill-behavior and serves as a challenging benchmark to evaluate downstream tasks.

## Acknowledgments and Disclosure of Funding

We gratefully acknowledge Jesse Krijthe, Taylan Turan, and the anonymous reviewers for their valuable reviews and insightful suggestions. We also thank Jan van Gemert, Marcel Reinders, David Tax, and Gijs van Tulder for their constructive discussions. We further acknowledge the support of DAIC (Delft AI Cluster) and DelftBlue for providing computational resources. Moreover, we thank the community, particularly OpenML for enabling the sharing of data and Scikit-learn for providing essential machine learning tools.

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

# A    LCDB 1.1 Additional Details

**Data Splitting.**    We split the data twice, first, the *outer split* (outer seed) splits off 10% test data. The *inner split* (inner seed) splits the remainder in a train (90%) and validation (10%) set. Validation and testing set are capped at 5000 samples. We use 5 inner and 5 outer seeds and these splits are stratified. Training sets are further reduced in size to simulate the collection of a learning curve. The training sets are constructed in a monotonic way without stratification, i.e. $S_1 \subset S_2 \subset ... \subset S_n$. This procedure corresponds exactly with how the LCDB 1.0 also was collected [10].

**Imputation.**    We impute the median for numerical features and the most frequent value for categorical features to deal with missing data. For categorical features, we apply one-hot encoding. This procedure corresponds exactly with how the LCDB 1.0 also was collected [10]. Note that, for LCDB 1.0, if the number of features is very large, features were binarized, which we believe was not a intended preprocessing step. We do not include any binarization.

**Justification of Other Additional Learners.**    The Complement Naive Bayes learner was introduced to resolve poor assumptions of the Multinomial Naive Bayes classifier, hence we include it [76]. Complement and Multinomial Naive Bayes are intended for text classification, yet few datasets are text datasets. Therefore, we decided to also include Gaussian Naive Bayes, which assumes features are Gaussian, which can be more reasonable for our diversity of datasets. We, however, choose to include all Naive Bayes learners, as they were also included in the LCDB 1.0. The Nearest Centroid classifier is computationally efficient but is known to display ill-behavior in toy settings [35].

**Naive Bayes Preprocessing and Mix-Naive Bayes.**    Each Naive Bayes model is included twice: as an original and mixed version. In LCDB 1.0, Naive Bayes was trained on all features, including the one-hot encoded features, which we call original. One-hot encoded features violate the core assumption of conditional independence that underlies the Naive Bayes model [68]. The mixed Naive Bayes models categorical and numerical features separately. Categorical Naive Bayes is used for categorical features, and the other model is used on the numerical features (Bernoulli, Multinomial, Complement, Guassian), ensuring that the categorical features are modeled appropriately.

**Modern Learner Implementation Details.**    We use the official implementations of CatBoost, TabNet, RealMLP, and TabPFN v2 with their default hyperparameters. For TabNet, we employ the small-scale model (TabNet-S) and use the default hyperparameters without early stopping, different from how TabNet was configured in [23], to ensure consistency across all learners.

# B    Difference Between LCDB 1.1 versions

To assess whether data leakage meaningfully alters the learning curves, we computed the proportion of instances where a statistically significant difference (based on Bonferroni-corrected $t$-tests) was observed between results obtained with and without potential leakage. This comparison was performed across three preprocessing configurations: no feature scaling, min-max normalization, and standardization. If there is one anchor significantly different between two curves, we classify it as a different curve. The results are summarized in Table 5. Observe that, for the case of no feature scaling, the amount of different curves is small, because here only imputation was performed. When feature scaling is used, data leakage becomes more pronounced.

Table 5: Percentage of curves with at least one anchor that is significantly different between data leakage and no data leakage version of the LCDB 1.1.

|  | no FS | min-max FS | standardization FS |
|---|---|---|---|
| LCDB 1.1 CC-18 (72) | 1.2% | 12.3% | 8.3% |
| LCDB 1.1 FULL (265) | 1.9% | 15.6% | 12.5% |

# C   Absolute Performance Comparison

In addition to analyzing the shape of the learning curve, we can also compare the performance of different learning algorithms on different feature scaling techniques by using optimal performance points on the learning curve. As a simple supplementary analysis provided in the appendix, Figure 7 presents the best average error rates of different learners on both LCDB 1.1 CC-18 and FULL. Specifically, we extract the minimum error rate from each learning curve and compute the average across datasets for each learner. It is evident that feature scaling has minimal impact on tree-based algorithms, but can significantly improve the performance of many distance-based and iteratively fitted learners.

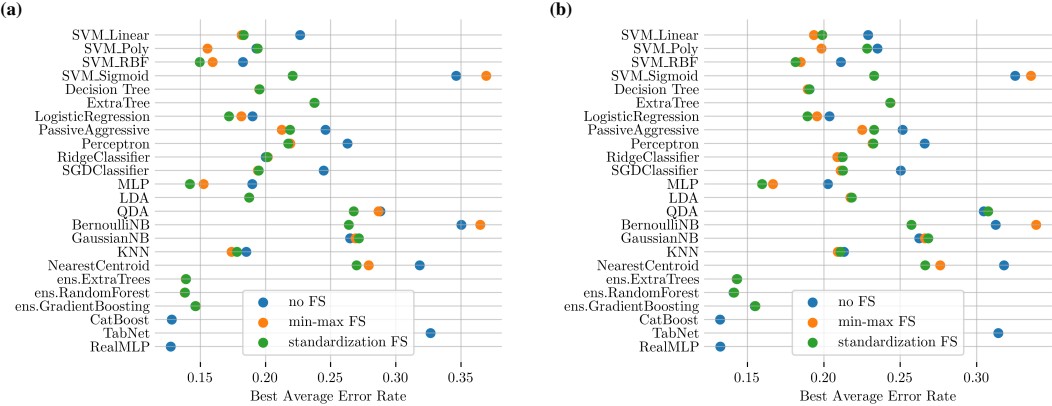

Figure 7: Comparison of learner best performance on average under different feature scaling strategies. (a) LCDB 1.1 CC-18 (72 datasets). (b) LCDB 1.1 FULL (265 datasets).

Furthermore, we can investigate the learners' optimal performance by filtering datasets according to their characteristics. This analysis can serve as a simple use case that provides further evidence on whether deep learning models outperform tree-based methods under different scale of dataset. Figure 8 compares the learners' performance between two groups: one where at least one anchor includes more than 10k training samples, and another where all anchors have fewer than 10k training samples. As shown, CatBoost and RealMLP demonstrate consistently strong performance across both groups, while TabNet exhibits competitive performance only on datasets with larger training set sizes.

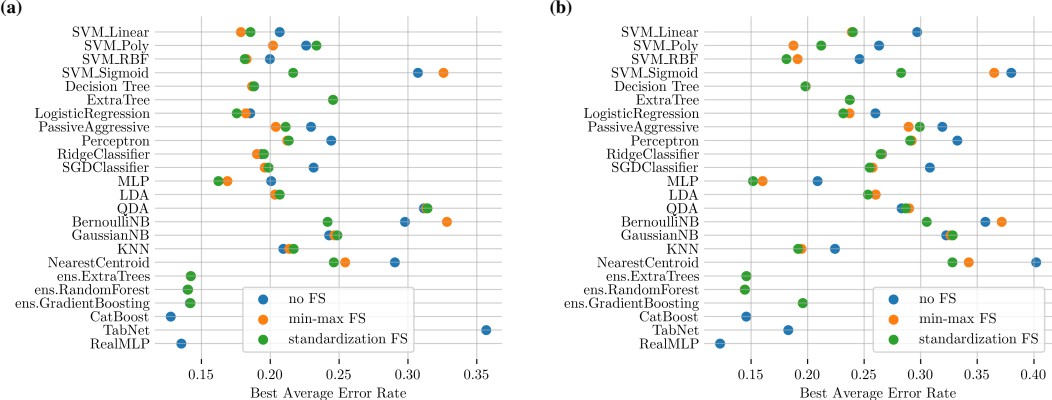

Figure 8: Comparison of learner best performance on average under different feature scaling strategies. (a) LCDB 1.1 FULL (200 datasets with number of samples less than 10k). (b) LCDB 1.1 FULL (65 datasets with number of samples more than 10k).

Motivated by the analysis in Williams [68], which highlights the incorrect assumption of conditional independence in Naive Bayes when applied to one-hot encoded features, we explore mixed Naive Bayes model variants. The one-hot encoded features are not independent and may lead to inaccurate

probability estimates and model misspeficiation. To address this, we introduce mixed Naive Bayes models and evaluate their performance using the win-loss-tie framework on the LCDB 1.1 FULL. We compare the performances over all anchors and datasets, and record a win if one method is better than the other and a tie if they achieve the same performance. As shown in Figure 9, mixed Naive Bayes models do not always outperform their vanilla counterparts. The tie cases are primarily due to datasets without categorical features, where no encoding is applied and both models behave identically. In conclusion, the model-misspecification for Naive Bayes does not seem so problematic for the error rate.

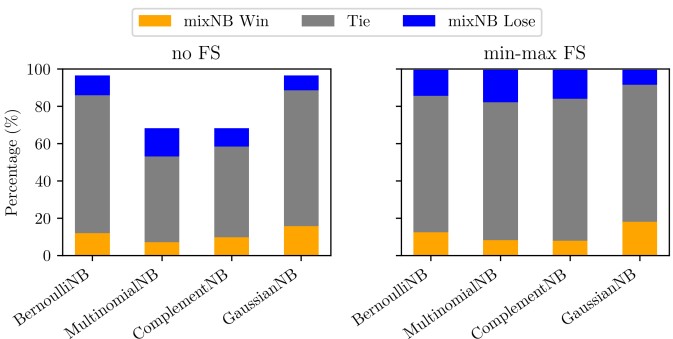

Figure 9: The performance comparison between Naive Bayes and mixed Naive Bayes methods by using LCDB 1.1 FULL (265 OpenML datasets).

# D Statistics in LCDB 1.1 per Learner

Similar to Table 4, we show the statistics of different learners in LCDB 1.1 for both CC-18 and FULL with no data-leakage version in Tables 6, 7, and 8. Here, non-monotone, non-convex, and ill-behaved refer to shapes that violate monotonicity, convexity, and either, respectively. In Table 6, we only include the no feature scaling version, since the three considered models can internally handle feature scaling. Note, some datasets from CC-18 or FULL were used in the meta-train benchmark for designing and meta-tuning RealMLP. For Table 7, the relatively high proportion of missing curves is mainly due to the limitations of TabPFN v2, which only supports datasets with up to 10k training samples, 500 features, and 10 classes, as well as curves whose maximum length is below 10k. Moreover, some datasets included in CC-18 and FULL were used during the design of the TabPFN prior. Therefore, the comparison involving TabPFN v2 is not entirely fair, and we present this table separately.

Table 6: Statistics of CatBoost, TabNet, and RealMLP in LCDB 1.1 (no feature scaling).

| Learner / Ratio(%) | LCDB 1.1 CC-18 (72) | | | | | | LCDB 1.1 FULL (265) | | | | | |
|---|---|---|---|---|---|---|---|---|---|---|---|---|
| | Missing | Non-Monotone | Non-Convex | Ill-behaved | Peaking | Dipping | Missing | Non-Monotone | Non-Convex | Ill-behaved | Peaking | Dipping |
| CatBoost | 0.0% | 0.0% | 0.0% | 0.0% | 0.0% | 0.0% | 0.0% | 0.4% | 1.1% | 1.5% | 0.8% | 0.4% |
| TabNet | 0.0% | 11.1% | 72.2% | 72.2% | 33.3% | 1.4% | 0.4% | 17.4% | 73.6% | 74.3% | 42.3% | 4.2% |
| RealMLP | 0.0% | 0.0% | 1.4% | 1.4% | 0.0% | 0.0% | 0.0% | 0.4% | 4.9% | 5.3% | 0.0% | 0.0% |

Table 7: Statistics of TabPFN v2 in LCDB 1.1.

| Learner / Ratio(%) | LCDB 1.1 CC-18 (72) | | | LCDB 1.1 FULL (265) | | |
|---|---|---|---|---|---|---|
| | no FS | min-max FS | standardization FS | no FS | min-max FS | standardization FS |
| Missing | 12.5% | 12.5% | 12.5% | 20.8% | 20.8% | 20.8% |
| Non-Monotone | 0.0% | 1.4% | 0.0% | 0.0% | 0.4% | 0.4% |
| Non-Convex | 0.0% | 1.4% | 0.0% | 1.5% | 2.3% | 1.9% |
| Ill-behaved | 0.0% | 2.8% | 0.0% | 1.5% | 2.6% | 1.9% |
| Peaking | 0.0% | 0.0% | 0.0% | 0.4% | 0.4% | 0.4% |
| Dipping | 0.0% | 0.0% | 0.0% | 0.0% | 0.0% | 0.0% |

Table 8: Statistics of 24 learners in LCDB 1.1.

**LCDB 1.1 CC-18 (72) no FS**

| Learner / Ratio(%) | Missing | Flat | Non-Monotone | Non-Convex | Ill-behaved | Peaking | Dipping |
|---|---|---|---|---|---|---|---|
| SVM_Linear | 0.0 | 3.4 | 4.2 | 4.2 | 5.6 | 2.8 | 2.8 |
| SVM_Poly | 0.0 | 19.4 | 0.0 | 4.2 | 4.2 | 2.8 | 0.0 |
| SVM_RBF | 0.0 | 19.4 | 1.4 | 11.1 | 11.1 | 6.9 | 1.4 |
| SVM_Sigmoid | 0.0 | 11.1 | 61.1 | 33.3 | 65.3 | 2.8 | 56.9 |
| Decision Tree | 0.0 | 2.8 | 0.0 | 0.0 | 0.0 | 0.0 | 0.0 |
| ExtraTree | 0.0 | 2.8 | 1.4 | 0.0 | 1.4 | 0.0 | 0.0 |
| LogisticRegression | 0.0 | 2.8 | 0.0 | 4.2 | 4.2 | 0.0 | 0.0 |
| PassiveAggressive | 0.0 | 1.4 | 5.6 | 0.0 | 5.6 | 1.4 | 1.4 |
| Perceptron | 0.0 | 0.0 | 4.2 | 1.4 | 4.2 | 13.9 | 4.2 |
| RidgeClassifier | 0.0 | 5.6 | 18.1 | 20.8 | 20.8 | 0.0 | 2.8 |
| SGDClassifier | 0.0 | 0.0 | 2.8 | 0.0 | 2.8 | 8.3 | 2.8 |
| MLP | 0.0 | 2.8 | 18.1 | 26.4 | 29.2 | 37.5 | 1.4 |
| LDA | 0.0 | 2.8 | 43.1 | 47.2 | 48.6 | 12.5 | 30.6 |
| QDA | 0.0 | 2.8 | 44.4 | 40.3 | 52.8 | 8.3 | 11.1 |
| BernoulliNB | 0.0 | 19.4 | 13.9 | 19.4 | 26.4 | 1.4 | 4.2 |
| MultinomialNB | 22.2 | 2.8 | 1.4 | 2.8 | 4.2 | 0.0 | 5.6 |
| ComplementNB | 22.2 | 2.8 | 6.9 | 2.8 | 8.3 | 2.8 | 19.4 |
| GaussianNB | 0.0 | 2.8 | 18.1 | 13.9 | 20.8 | 0.0 | 8.3 |
| KNN | 0.0 | 13.9 | 4.2 | 2.8 | 5.6 | 0.0 | 5.6 |
| NearestCentroid | 0.0 | 4.2 | 4.2 | 2.8 | 5.6 | 1.4 | 5.6 |
| ens.ExtraTrees | 0.0 | 8.3 | 1.4 | 0.0 | 1.4 | 0.0 | 1.4 |
| ens.RandomForest | 0.0 | 6.9 | 1.4 | 0.0 | 1.4 | 0.0 | 1.4 |
| ens.GradientBoosting | 0.0 | 0.0 | 0.0 | 0.0 | 0.0 | 0.0 | 0.0 |
| DummyClassifier | 0.0 | 73.6 | 15.3 | 65.3 | 69.4 | 62.5 | 2.8 |

**LCDB 1.1 CC-18 (72) min-max FS**

| Learner / Ratio(%) | Missing | Flat | Non-Monotone | Non-Convex | Ill-behaved | Peaking | Dipping |
|---|---|---|---|---|---|---|---|
| SVM_Linear | 0.0 | 5.6 | 1.4 | 0.0 | 1.4 | 0.0 | 0.0 |
| SVM_Poly | 0.0 | 2.8 | 0.0 | 1.4 | 1.4 | 0.0 | 0.0 |
| SVM_RBF | 0.0 | 15.3 | 8.3 | 8.3 | 8.3 | 6.9 | 1.4 |
| SVM_Sigmoid | 0.0 | 11.1 | 58.3 | 33.3 | 62.5 | 4.2 | 56.9 |
| Decision Tree | 0.0 | 0.0 | 1.4 | 0.0 | 1.4 | 0.0 | 0.0 |
| ExtraTree | 0.0 | 13.9 | 0.0 | 0.0 | 0.0 | 0.0 | 0.0 |
| LogisticRegression | 0.0 | 1.4 | 0.0 | 0.0 | 0.0 | 0.0 | 2.8 |
| PassiveAggressive | 0.0 | 0.0 | 0.0 | 0.0 | 0.0 | 0.0 | 0.0 |
| Perceptron | 0.0 | 11.1 | 1.4 | 5.6 | 5.6 | 1.4 | 1.4 |
| RidgeClassifier | 0.0 | 0.0 | 0.0 | 0.0 | 0.0 | 0.0 | 4.2 |
| SGDClassifier | 0.0 | 2.8 | 0.0 | 5.6 | 5.6 | 8.3 | 2.8 |
| MLP | 0.0 | 6.9 | 16.7 | 18.1 | 19.4 | 37.5 | 1.4 |
| LDA | 0.0 | 2.8 | 43.1 | 47.2 | 48.6 | 13.9 | 33.3 |
| QDA | 0.0 | 13.9 | 47.2 | 50.0 | 59.7 | 0.0 | 19.4 |
| BernoulliNB | 0.0 | 12.5 | 18.1 | 9.7 | 20.8 | 1.4 | 9.7 |
| MultinomialNB | 100.0 | 0.0 | 6.9 | 5.6 | 12.5 | 1.4 | 15.3 |
| ComplementNB | 100.0 | 0.0 | 15.3 | 11.1 | 18.1 | 6.9 | 22.2 |
| GaussianNB | 0.0 | 1.4 | 23.6 | 19.4 | 25.0 | 6.9 | 8.3 |
| KNN | 0.0 | 11.1 | 2.8 | 2.8 | 4.2 | 0.0 | 19.4 |
| NearestCentroid | 0.0 | 4.2 | 18.1 | 4.2 | 18.1 | 0.0 | 5.6 |
| ens.ExtraTrees | 0.0 | 6.9 | 1.4 | 0.0 | 1.4 | 0.0 | 1.4 |
| ens.RandomForest | 0.0 | 6.9 | 1.4 | 0.0 | 1.4 | 0.0 | 1.4 |
| ens.GradientBoosting | 0.0 | 0.0 | 0.0 | 0.0 | 0.0 | 0.0 | 0.0 |
| DummyClassifier | 0.0 | 73.6 | 15.3 | 65.3 | 69.4 | 62.5 | 2.8 |

**LCDB 1.1 CC-18 (72) standardization FS**

| Learner / Ratio(%) | Missing | Flat | Non-Monotone | Non-Convex | Ill-behaved | Peaking | Dipping |
|---|---|---|---|---|---|---|---|
| SVM_Linear | 0.0 | 1.4 | 0.0 | 1.4 | 1.4 | 0.0 | 0.0 |
| SVM_Poly | 0.0 | 6.9 | 0.0 | 4.2 | 4.2 | 0.0 | 0.0 |
| SVM_RBF | 0.0 | 12.5 | 0.0 | 4.2 | 4.2 | 0.0 | 0.0 |
| SVM_Sigmoid | 0.0 | 6.9 | 44.4 | 23.6 | 47.2 | 0.0 | 54.2 |
| Decision Tree | 0.0 | 0.0 | 1.4 | 0.0 | 1.4 | 0.0 | 1.4 |
| ExtraTree | 0.0 | 0.0 | 1.4 | 0.0 | 1.4 | 0.0 | 0.0 |
| LogisticRegression | 0.0 | 4.2 | 0.0 | 2.8 | 2.8 | 0.0 | 0.0 |
| PassiveAggressive | 0.0 | 0.0 | 1.4 | 0.0 | 1.4 | 0.0 | 1.4 |
| Perceptron | 0.0 | 0.0 | 16.7 | 18.1 | 18.1 | 12.5 | 1.4 |
| RidgeClassifier | 0.0 | 4.2 | 2.8 | 2.8 | 4.2 | 0.0 | 1.4 |
| SGDClassifier | 0.0 | 2.8 | 5.6 | 5.6 | 6.9 | 0.0 | 2.8 |
| MLP | 0.0 | 1.4 | 44.4 | 48.6 | 51.4 | 37.5 | 1.4 |
| LDA | 0.0 | 0.0 | 47.2 | 48.6 | 58.3 | 9.7 | 31.9 |
| QDA | 0.0 | 6.9 | 4.2 | 5.6 | 6.9 | 0.0 | 5.6 |
| BernoulliNB | 0.0 | 0.0 | 0.0 | 0.0 | 0.0 | 0.0 | 0.0 |
| MultinomialNB | 100.0 | 0.0 | 0.0 | 0.0 | 0.0 | 0.0 | 0.0 |
| ComplementNB | 100.0 | 1.4 | 26.4 | 22.2 | 29.2 | 6.9 | 25.0 |
| GaussianNB | 0.0 | 2.8 | 6.9 | 2.8 | 6.9 | 6.9 | 5.6 |
| KNN | 0.0 | 12.5 | 6.9 | 1.4 | 6.9 | 0.0 | 11.1 |
| NearestCentroid | 0.0 | 2.8 | 6.9 | 1.4 | 6.9 | 0.0 | 1.4 |
| ens.ExtraTrees | 0.0 | 6.9 | 1.4 | 0.0 | 1.4 | 0.0 | 1.4 |
| ens.RandomForest | 0.0 | 6.9 | 1.4 | 0.0 | 1.4 | 0.0 | 1.4 |
| ens.GradientBoosting | 0.0 | 0.0 | 0.0 | 0.0 | 0.0 | 0.0 | 0.0 |
| DummyClassifier | 0.0 | 73.6 | 15.3 | 65.3 | 69.4 | 62.5 | 2.8 |

**LCDB 1.1 FULL (265) no FS**

| Learner / Ratio(%) | Missing | Flat | Non-Monotone | Non-Convex | Ill-behaved | Peaking | Dipping |
|---|---|---|---|---|---|---|---|
| SVM_Linear | 0.0 | 3.4 | 3.8 | 4.5 | 5.7 | 2.3 | 2.6 |
| SVM_Poly | 0.0 | 17.0 | 3.4 | 6.4 | 7.9 | 3.8 | 3.4 |
| SVM_RBF | 0.0 | 18.1 | 3.8 | 15.5 | 16.2 | 7.2 | 2.3 |
| SVM_Sigmoid | 0.0 | 19.2 | 48.3 | 37.7 | 58.5 | 7.9 | 46.4 |
| Decision Tree | 0.0 | 3.8 | 0.8 | 1.5 | 1.5 | 0.8 | 1.5 |
| ExtraTree | 0.0 | 4.5 | 1.9 | 0.4 | 1.9 | 0.0 | 1.1 |
| LogisticRegression | 0.0 | 6.8 | 2.3 | 4.5 | 5.3 | 1.5 | 1.9 |
| PassiveAggressive | 0.0 | 5.7 | 8.3 | 4.9 | 9.4 | 2.3 | 3.8 |
| Perceptron | 0.0 | 7.2 | 3.0 | 2.3 | 3.8 | 1.1 | 1.9 |
| RidgeClassifier | 0.0 | 2.3 | 14.7 | 15.1 | 17.0 | 10.6 | 5.3 |
| SGDClassifier | 0.0 | 4.9 | 3.0 | 1.1 | 3.4 | 0.8 | 3.0 |
| MLP | 0.0 | 3.8 | 21.1 | 21.9 | 27.9 | 8.3 | 4.2 |
| LDA | 0.0 | 3.8 | 32.5 | 33.2 | 37.7 | 24.5 | 7.2 |
| QDA | 0.0 | 26.4 | 34.0 | 37.4 | 46.0 | 12.1 | 27.5 |
| BernoulliNB | 30.2 | 9.1 | 12.1 | 23.4 | 28.7 | 13.6 | 8.3 |
| MultinomialNB | 30.2 | 8.3 | 4.9 | 7.2 | 8.3 | 2.3 | 6.0 |
| ComplementNB | 0.0 | 4.5 | 6.8 | 5.7 | 8.3 | 1.5 | 7.5 |
| GaussianNB | 0.0 | 23.0 | 23.0 | 15.8 | 25.3 | 4.5 | 25.7 |
| KNN | 0.0 | 10.9 | 2.6 | 1.9 | 3.8 | 0.8 | 4.5 |
| NearestCentroid | 0.0 | 10.9 | 7.5 | 5.3 | 8.3 | 1.5 | 13.2 |
| ens.ExtraTrees | 0.0 | 9.1 | 1.9 | 2.3 | 3.4 | 1.1 | 1.9 |
| ens.RandomForest | 0.0 | 9.1 | 1.5 | 1.9 | 3.0 | 1.1 | 1.5 |
| ens.GradientBoosting | 0.0 | 3.4 | 1.1 | 0.8 | 1.9 | 0.4 | 0.8 |
| DummyClassifier | 0.0 | 69.4 | 16.6 | 54.7 | 60.4 | 49.1 | 3.8 |

**LCDB 1.1 FULL (265) min-max FS**

| Learner / Ratio(%) | Missing | Flat | Non-Monotone | Non-Convex | Ill-behaved | Peaking | Dipping |
|---|---|---|---|---|---|---|---|
| SVM_Linear | 0.0 | 7.9 | 2.6 | 2.3 | 3.4 | 2.3 | 2.6 |
| SVM_Poly | 0.0 | 5.7 | 1.5 | 2.6 | 3.8 | 3.8 | 3.4 |
| SVM_RBF | 0.0 | 13.2 | 3.0 | 9.8 | 11.3 | 7.2 | 2.3 |
| SVM_Sigmoid | 0.0 | 20.0 | 44.5 | 34.7 | 55.8 | 7.9 | 46.4 |
| Decision Tree | 0.0 | 2.3 | 1.5 | 1.5 | 2.3 | 0.8 | 1.5 |
| ExtraTree | 0.0 | 2.3 | 1.9 | 0.4 | 1.9 | 0.0 | 1.1 |
| LogisticRegression | 0.0 | 11.3 | 1.5 | 1.5 | 1.9 | 1.5 | 1.9 |
| PassiveAggressive | 0.0 | 5.3 | 2.3 | 0.8 | 2.6 | 0.8 | 3.8 |
| Perceptron | 0.0 | 1.9 | 1.5 | 3.8 | 4.2 | 1.9 | 1.9 |
| RidgeClassifier | 0.0 | 10.9 | 2.3 | 1.1 | 1.5 | 6.4 | 1.1 |
| SGDClassifier | 0.0 | 3.0 | 1.1 | 23.0 | 26.8 | 24.5 | 2.3 |
| MLP | 0.0 | 7.5 | 18.5 | 33.2 | 37.7 | 24.5 | 7.2 |
| LDA | 0.0 | 4.2 | 32.5 | 40.8 | 48.7 | 12.1 | 27.5 |
| QDA | 0.0 | 22.3 | 35.1 | 12.5 | 23.4 | 2.6 | 8.3 |
| BernoulliNB | 0.0 | 14.3 | 17.7 | 9.8 | 18.9 | 2.6 | 6.0 |
| MultinomialNB | 100.0 | 5.3 | 14.3 | 11.7 | 19.2 | 2.6 | 18.5 |
| ComplementNB | 100.0 | 4.5 | 17.7 | 20.0 | 29.4 | 8.7 | 26.4 |
| GaussianNB | 0.0 | 10.2 | 27.5 | 3.0 | 4.5 | 0.8 | 3.0 |
| KNN | 0.0 | 6.8 | 2.6 | 10.2 | 21.9 | 1.5 | 4.5 |
| NearestCentroid | 0.0 | 8.3 | 21.5 | 2.3 | 3.4 | 1.1 | 13.2 |
| ens.ExtraTrees | 0.0 | 8.3 | 1.9 | 2.3 | 3.4 | 1.1 | 1.9 |
| ens.RandomForest | 0.0 | 3.0 | 1.5 | 1.1 | 3.0 | 1.1 | 1.5 |
| ens.GradientBoosting | 0.0 | 3.0 | 1.1 | 0.8 | 1.9 | 0.4 | 0.8 |
| DummyClassifier | 0.0 | 69.4 | 16.6 | 54.7 | 60.4 | 49.1 | 3.8 |

**LCDB 1.1 FULL (265) standardization FS**

| Learner / Ratio(%) | Missing | Flat | Non-Monotone | Non-Convex | Ill-behaved | Peaking | Dipping |
|---|---|---|---|---|---|---|---|
| SVM_Linear | 0.0 | 2.3 | 3.4 | 4.2 | 5.3 | 1.9 | 2.3 |
| SVM_Poly | 0.0 | 12.8 | 2.3 | 5.3 | 6.0 | 2.3 | 1.9 |
| SVM_RBF | 0.0 | 12.1 | 3.4 | 6.0 | 7.2 | 1.5 | 0.8 |
| SVM_Sigmoid | 0.0 | 11.7 | 39.6 | 24.9 | 42.6 | 0.8 | 42.6 |
| Decision Tree | 0.0 | 3.0 | 1.9 | 1.5 | 2.3 | 0.8 | 1.5 |
| ExtraTree | 0.0 | 2.6 | 1.9 | 0.4 | 1.9 | 0.0 | 1.1 |
| LogisticRegression | 0.0 | 7.9 | 3.4 | 4.2 | 4.9 | 3.0 | 1.5 |
| PassiveAggressive | 0.0 | 1.9 | 4.2 | 1.9 | 4.9 | 1.1 | 3.0 |
| Perceptron | 0.0 | 1.1 | 3.0 | 1.9 | 3.0 | 1.1 | 2.3 |
| RidgeClassifier | 0.0 | 6.0 | 14.7 | 15.8 | 17.7 | 10.2 | 2.6 |
| SGDClassifier | 0.0 | 0.8 | 3.4 | 3.0 | 4.2 | 1.5 | 1.9 |
| MLP | 0.0 | 3.0 | 9.4 | 14.0 | 16.6 | 3.0 | 7.2 |
| LDA | 0.0 | 1.9 | 32.1 | 33.6 | 37.7 | 23.8 | 34.0 |
| QDA | 0.0 | 9.1 | 38.1 | 40.0 | 47.5 | 9.8 | 9.4 |
| BernoulliNB | 0.0 | 0.0 | 8.3 | 8.3 | 12.1 | 1.1 | 0.0 |
| MultinomialNB | 100.0 | 0.0 | 0.0 | 0.0 | 0.0 | 0.0 | 0.0 |
| ComplementNB | 100.0 | 0.0 | 0.0 | 0.0 | 0.0 | 0.0 | 0.0 |
| GaussianNB | 0.0 | 3.4 | 28.3 | 21.1 | 30.6 | 8.3 | 27.5 |
| KNN | 0.0 | 10.9 | 5.7 | 3.4 | 6.8 | 0.8 | 4.5 |
| NearestCentroid | 0.0 | 8.7 | 12.8 | 7.5 | 13.2 | 1.1 | 15.5 |
| ens.ExtraTrees | 0.0 | 8.3 | 1.9 | 2.3 | 3.4 | 1.1 | 1.9 |
| ens.RandomForest | 0.0 | 3.0 | 1.5 | 2.3 | 3.0 | 1.1 | 1.1 |
| ens.GradientBoosting | 0.0 | 3.0 | 0.8 | 0.4 | 1.1 | 0.4 | 0.4 |
| DummyClassifier | 0.0 | 69.8 | 16.6 | 54.7 | 60.4 | 49.1 | 3.8 |

# E Observation the Shape of Learning Curves

In this section, we provide additional details beyond the detection of ill-behaviors and present some results of more measurements. Specifically, we include some analyses on the localization of peakings in MLP and LDA with different feature scaling settings. Meanwhile, we visualize some results about the size of monotonicity and convexity violations (violation errors), which quantifies the severity of such behaviors. In addition, we present further learner-wise analyses related to the shape of learning curves, including the standard deviation across different random seeds. These results aim to provide deeper insights into how models behave learning differently and also show a new perspective on how our LCDB 1.1 can be used.

## E.1 Surprising Shapes of MLP

In Figure 10, we show the cases where the MLP exhibits a phase transition in LCDB 1.1 CC-18. Since identifying such patterns, which characterized by abrupt improvements in performance, is somewhat subjective and they occur relatively infrequently, we did not develop a method to detect them. Instead, we selected them manually and observed that these transitions can consistently be eliminated through feature scaling.

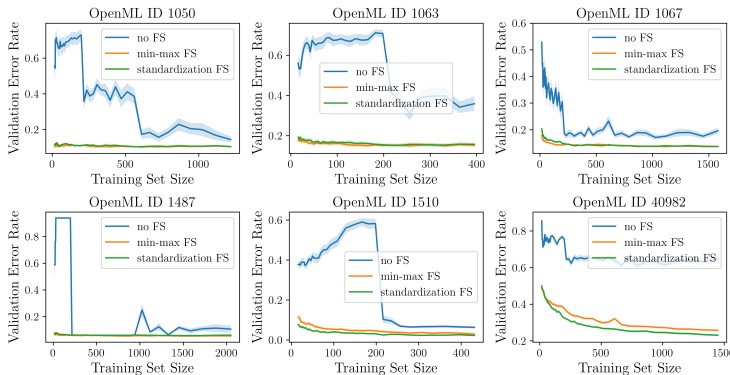

Figure 10: The phase transition shapes of MLP in LCDB 1.1 CC-18.

In Figure 11, we visualize the location of the peak for the MLP. Specifically, the peaking detection process involves a convexity violation analysis. We identify the point of maximum convexity violation (definition in Eq. 3) and extract the coordinates of its middle point ($i^*$), which we take as the estimated peak position.

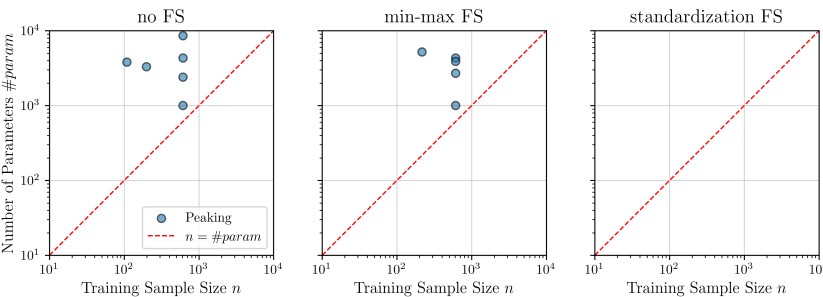

Figure 11: The position of peaks for the MLP for different feature scalings. Standard scaling seems to completely resolve peaking.

The location of the peaks line up suspiciously vertically. We discovered that the peak location can be explained as follows. When iterating over mini-batches, the last mini-batch can contain fewer samples because the training set sizes are not multiples of the mini-batch size. This can cause the last

mini-batch to contain one or very few samples. This causes convergence issues with fitting the MLP, leading to worse performance for very specific training set sizes. Notably, the peaks also disappear when standardization scaling, which is known to improve convergence for the SGD optimizer. Thus, the peaking behavior of the MLP is largely an issue due to how mini-batches are sampled in Scikit.

## E.2 Ill-Behaved Learning Curves in TabNet

The notably ill-behaved learner TabNet exhibits substantial non-convexity in its learning curves, which mostly appears to stem from phase transition phenomena. Figure 12 presents all the ill-behaved learning curves of TabNet in LCDB 1.1 CC-18. We can clearly observe that many of these curves show phase transition shapes, where the performance changes abruptly. From these observations, we find that such ill-behaviors typically occur at small training set sizes. As the training set size increases, the performance of TabNet tends to become more stable. In some cases, multiple phase transitions can even be observed within a single learning curve.

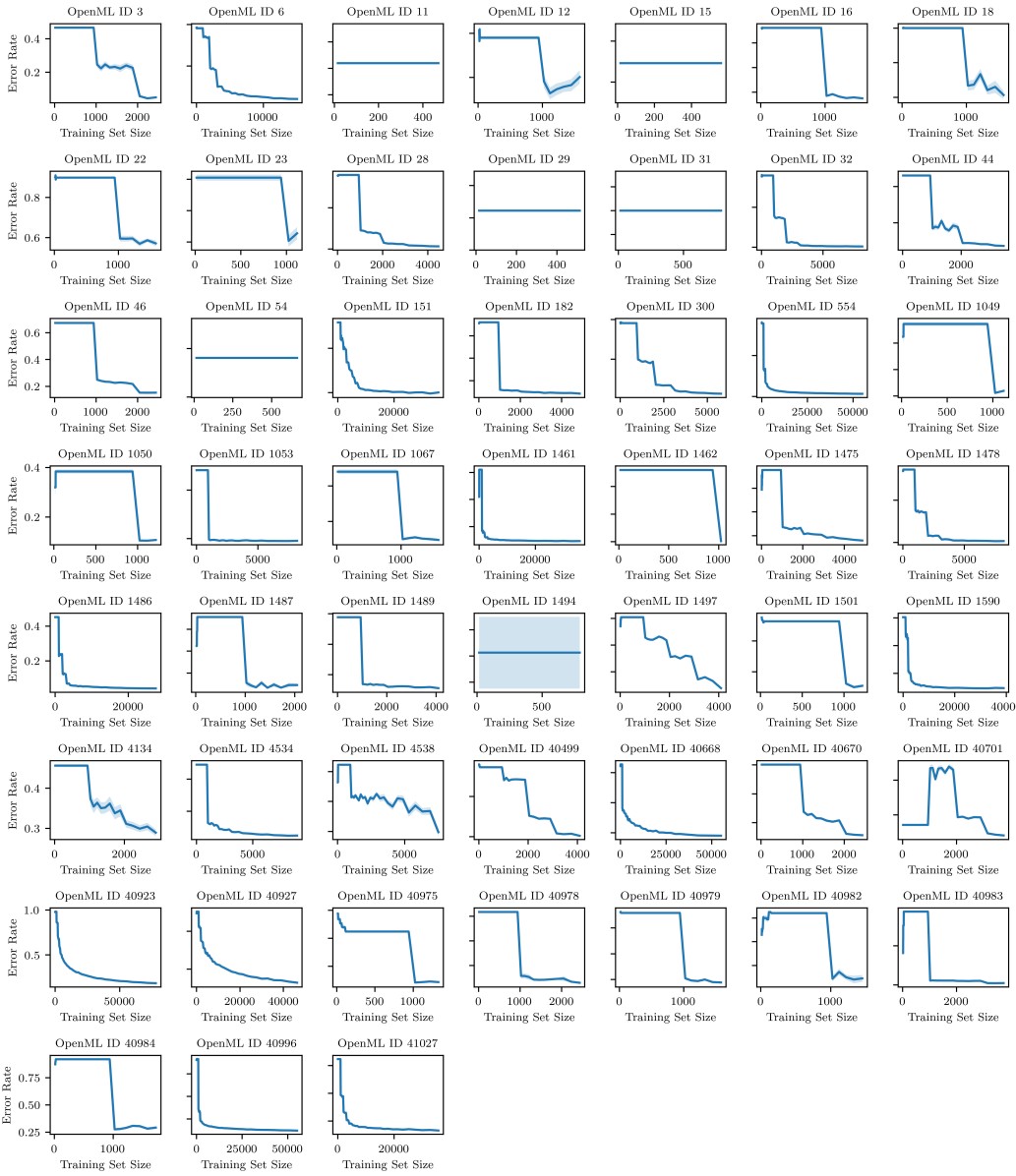

Figure 12: Ill-behaved learning curves of TabNet in LCDB 1.1 CC-18

### E.3 Standard Deviation Distribution of SVM

The performance of the model improves noticeably after scaling of features (Figure 13a). Furthermore, we observed substantial changes in the standard deviation over the repeats, indicating that the training process becomes more stable after feature scaling (Linear SVM as an example is shown in Figure 13b).

We extract the standard deviation of all anchors from all learning curves and, based on the frequency distribution, plot three histograms: one without feature scaling, one with min-max feature scaling, and one with standardization.

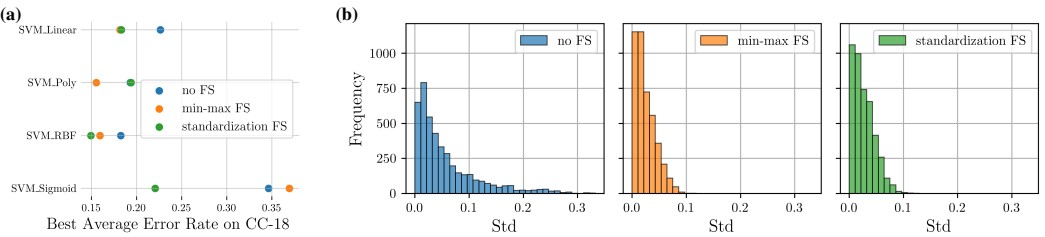

Figure 13: Feature scaling improve SVM performance and can make them more stable in the context of standard deviation. (a) Comparison of SVMs best error rate in average under different feature scaling. (b) Standard deviation distribution of learner Linear SVM, the Y-axis represents the times in all anchors of the learning curve.

### E.4 More Details regarding LDA

In Figure 14a we extract the location of the peak for LDA, and compare it to the dimensionality of the dataset. We see that the peak location occurs when the training set size is approximately equal to the dimensionality. Since LDA is insensitive to feature scaling, scaling the features does not lead to any changes in the learning curve (besides small differences due to numerical issues).

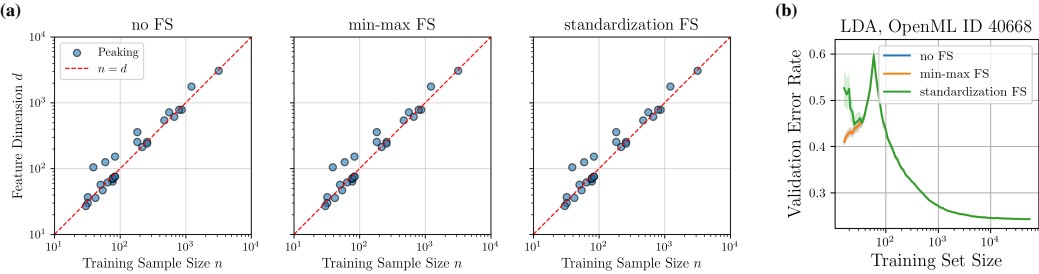

Figure 14: The curve shape of LDA is unaffected by feature scaling. (a) Peaking occurs when training set size is approximately equal to the dimensionality for LDA. (b) Numerical issues can cause a small change in the learning curve (this is the only curve in LCDB 1.1 that we can find that is different for LDA).

Another notable observation is that, although a substantial number of peaking behaviors in LDA learning curves are identified (see Section 5), a significant portion of peaks are still missed. Around 15% of the datasets in CC-18 contain fewer than 16 features, while the first anchor is defined at a training set size of 16. Given that $n \approx d$ in such cases, peaking behaviors are underestimated in the LDA learning curves.

To theoretically confirm that feature scaling does not affect the LDA learning curves, we provide the following proof.

*Proof.* Given the LDA discriminant function:

$$f_k(\mathbf{x}) = \mathbf{x}^T \mathbf{w}_k + w_{0k}, \tag{6}$$

where

$$\mathbf{w}_k = \mathbf{\Sigma}^{-1}\boldsymbol{\mu}_k, \tag{7}$$

$$w_{0k} = -\frac{1}{2}\boldsymbol{\mu}_k^T\mathbf{\Sigma}^{-1}\boldsymbol{\mu}_k + \log p(y_k). \tag{8}$$

$\mathbf{x}$ is the feature vector, $\boldsymbol{\mu}_k$ and $\mathbf{\Sigma}$ denote the class mean and shared covariance matrix, and $p(y_k)$ is the prior probability of class $k$.

The scaling factor for feature scaling transformation $\mathbf{S}$ is a diagonal matrix, after feature scaling $\mathbf{x}' = \mathbf{S}\mathbf{x}$. Correspondingly, the class means and covariance matrices become $\boldsymbol{\mu}'_k = \mathbf{S}\boldsymbol{\mu}_k$ and $\mathbf{\Sigma}' = \mathbf{S}\mathbf{\Sigma}\mathbf{S}$.

Then:

$$\begin{align}
\mathbf{w}'_k &= (\mathbf{\Sigma}')^{-1}\boldsymbol{\mu}'_k \tag{9}\\
&= (\mathbf{S}\mathbf{\Sigma}\mathbf{S})^{-1}(\mathbf{S}\boldsymbol{\mu}_k) \tag{10}\\
&= \mathbf{S}^{-1}\mathbf{\Sigma}^{-1}\boldsymbol{\mu}_k \tag{11}\\
&= \mathbf{S}^{-1}\mathbf{w}_k. \tag{12}
\end{align}$$

Similarly:

$$\begin{align}
w'_{0k} &= -\frac{1}{2}(\boldsymbol{\mu}'_k)^T(\mathbf{\Sigma}')^{-1}\boldsymbol{\mu}'_k + \log p(y_k) \tag{13}\\
&= -\frac{1}{2}(\mathbf{S}\boldsymbol{\mu}_k)^T(\mathbf{S}^{-1}\mathbf{\Sigma}^{-1}\mathbf{S}^{-1})\mathbf{S}\boldsymbol{\mu}_k + \log p(y_k) \tag{14}\\
&= -\frac{1}{2}\boldsymbol{\mu}_k^T\mathbf{\Sigma}^{-1}\boldsymbol{\mu}_k + \log p(y_k) \tag{15}\\
&= w_{0k}. \tag{16}
\end{align}$$

Thus, the new discriminant function:

$$\begin{align}
f'_k(\mathbf{x}') &= (\mathbf{w}'_k)^T\mathbf{x}' + w'_{0k} \tag{17}\\
&= (\mathbf{S}^{-1}\mathbf{w}_k)^T(\mathbf{S}\mathbf{x}) + w_{0k} \tag{18}\\
&= \mathbf{w}_k^T\mathbf{x} + w_{0k} \tag{19}
\end{align}$$

The new discriminant function is the same as no feature scaling one, so the learning curves should be exactly the same after feature scaling. $\qquad\square$

Note that a similar argument holds for QDA, which should also be insensitive to feature scaling. However, given the reproducibility issues of QDA we did not investigate empirically.

## E.5 Detailed Violations Error in CC-18

Since we define the violation error in Equations 2 and 3 to quantify the size of monotonicity and convexity violations, we are able to compare the severity of such violations across different learners and datasets. Figures 15 and 16 provide heatmap visualizations illustrating these violation errors, where zero means no violation, and white means missing learning curves.

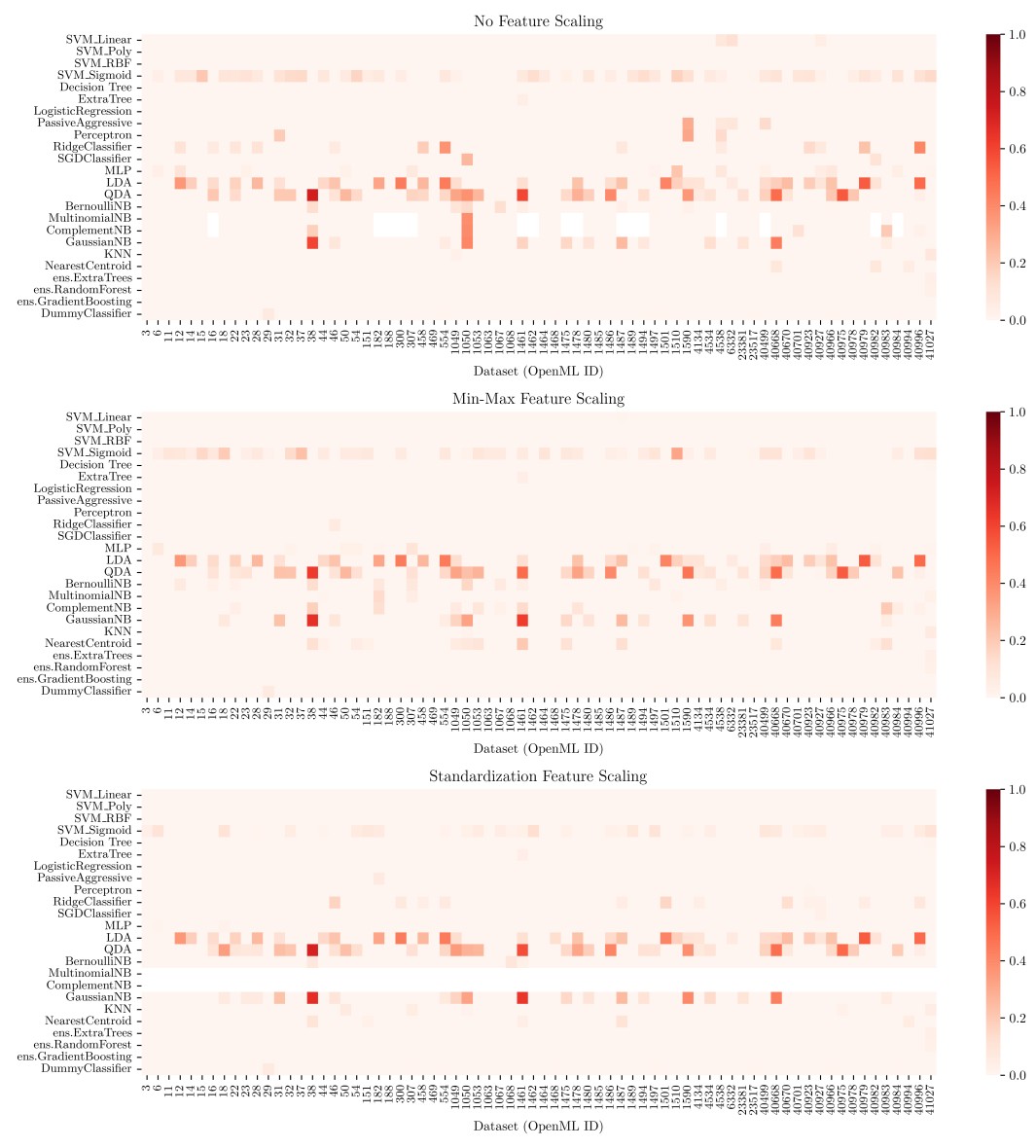

Figure 15: Monotonicity violations error heatmap of LCDB 1.1 CC-18.

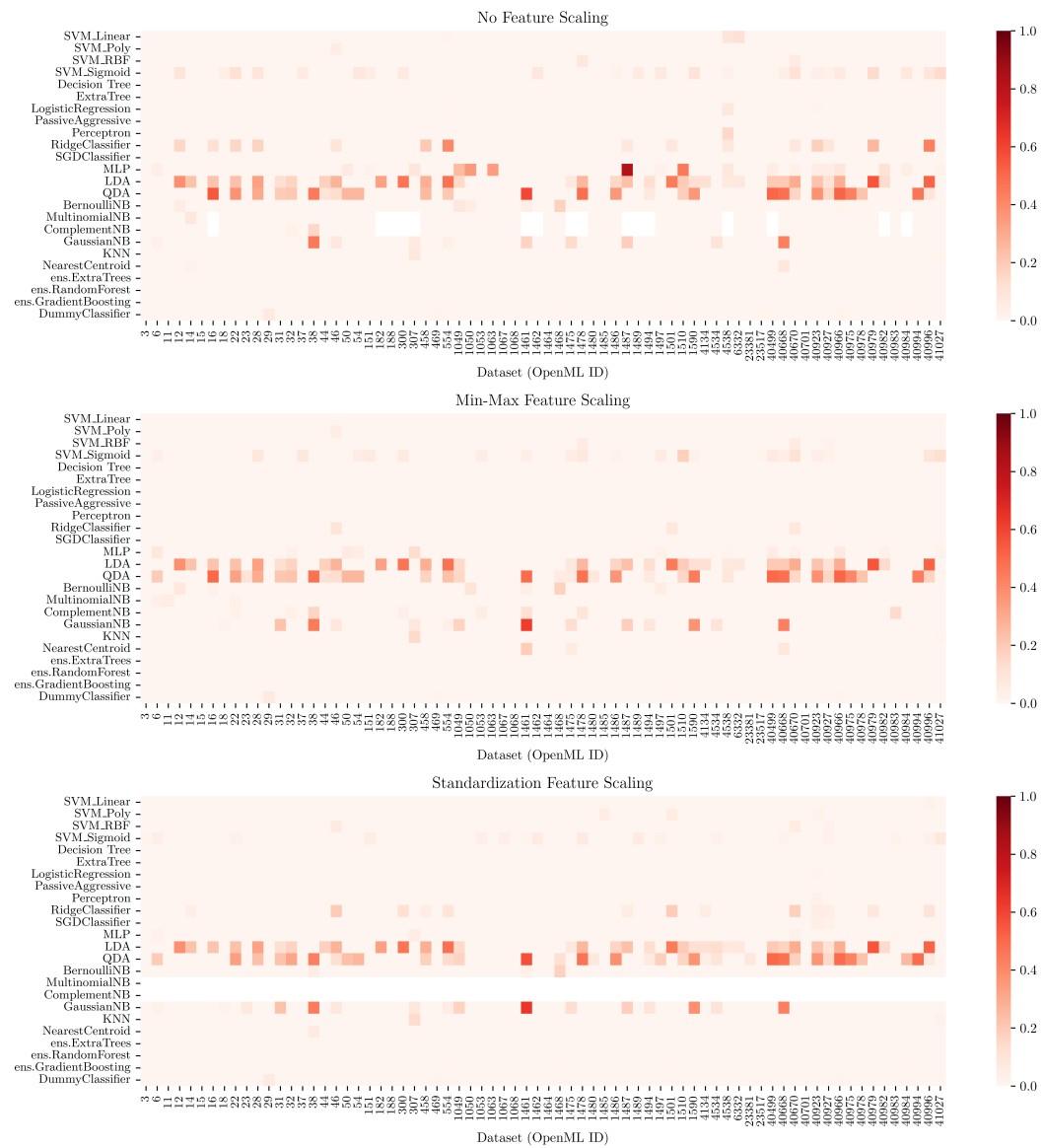

Figure 16: Convexity violations error heatmap of LCDB 1.1 CC-18.

# F  Learning Curves Fitting with More Parametric Models

To further validate the findings in Section 5.4, we include additional experiments with parametric models MMF4 and WBL4 by using the same experimental setting (Figure 17). These results confirm that the conclusions remain consistent.

The intrinsic properties of these parametric models conflict with the characteristics of some ill-behaved learning curves, which explains the observed experimental results. In particular, the phase transition shapes (only the convexity violated) can be effectively fitted by MMF4. The peaking and dipping, which violate the monotonicity, cannot be modeled by these parametric models. For reference, we also provide illustrative examples (Figure 18) highlighting the alignment (or misalignment) between model properties and the geometric characteristics of learning curves, as discussed in the main text.

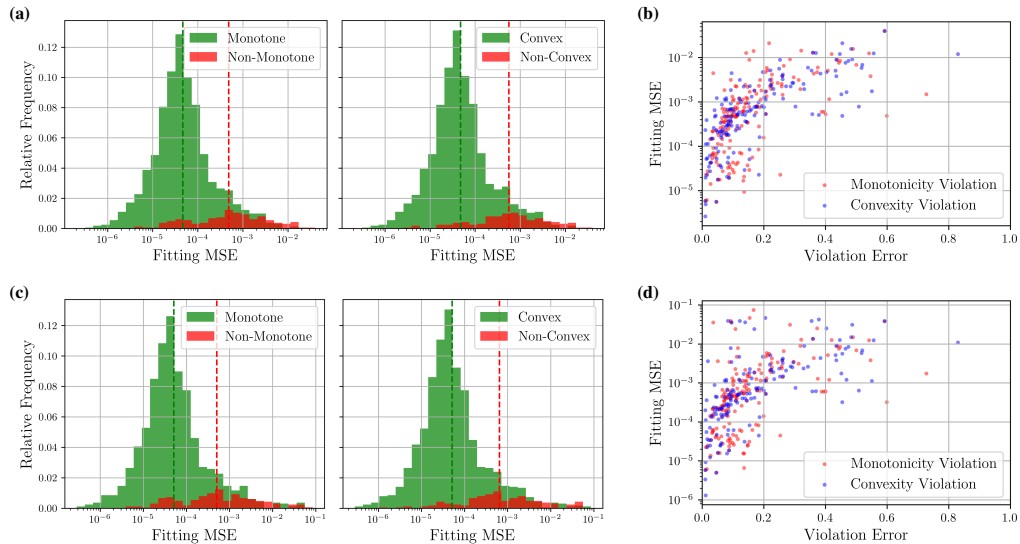

Figure 17: Ill-behaved (non-monotone or non-convex) learning curves are more difficult to fit with parametric models. MSE of the fitting is proportional to the violation error. (a, b) Models: MMF4 ($\frac{ab+cx^d}{b+x^d}$). (c, d) WBL4 ($-b\exp(-ax^d)+c$).

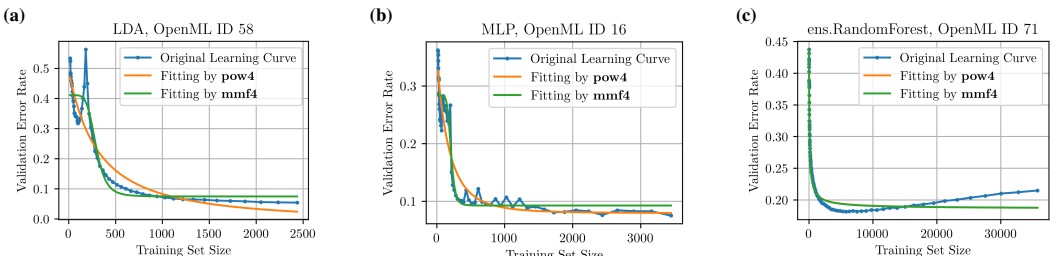

Figure 18: The parametric models fitting examples on ill-behaved shapes of learning curves. (a) Peaking. (b) Phase Transition. (c) Dipping.

## G   Learning Curve Crossings Affect Successive Halving

In this section, we provide detailed information about the experimental setup and results analyzing the relationship between learning curve crossings and the model selection performance of Successive Halving (SH) [61].

Figure 19 provides a pairwise crossing probability matrix for all learners to show that learning curves cross. The experiments are conducted in LCDB 1.1 CC-18 min-max feature scaling version since there are no missing curves. The left matrix shows the probability that learner A initially outperforms learner B at the lowest fidelity. The middle matrix refines this by showing the probability that A starts higher but ends lower than B, capturing the crossing from above. The right matrix shows the conditional probability of being overtaken given an early advantage, highlighting how frequently an initial lead fails to persist.

To showcase how we can use LCDB 1.1 to study the connection between SH model selection performance and crossing of learning curves, we show more detailed experimental results for comparing the performance of SH on two groups of learners, one with rarely and one with commonly crossing learning curves. In Figure 20, the blue and orange groups are the learners whose curves cross rarely and frequently, respectively. The left, middle, and right column figures are for the case where SH starts at the first available anchor (16 training instances), the 8th anchor (30 training instances), and the 16th anchor (59 training instances), respectively. From top to bottom, each row corresponds to a different per-round budget increase rate in the SH procedure, specifically 12.5%, 25.0%, 50.0%, and 100%. The per-round budget increase rate determines how much the training

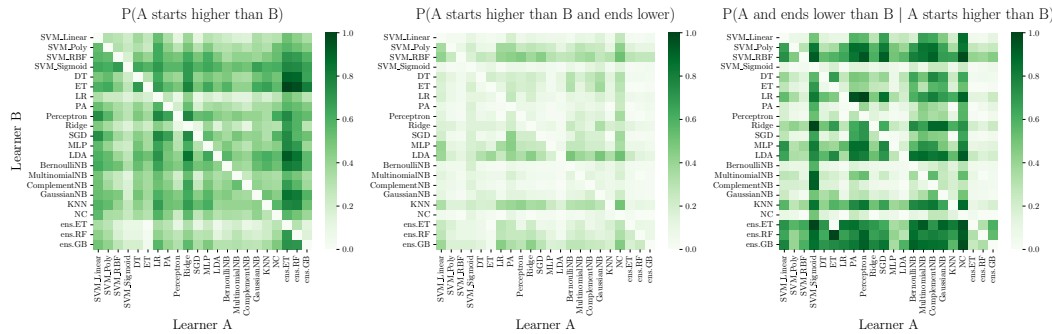

Figure 19: A simple method to evaluate crossing probability

budget is increased between consecutive SH rounds. The left panel of box-plots show, for different values of $k$, the probability (across the 5 outer folds of datasets) that the finally chosen algorithm is under the top $k$. However, since the final performance differences of learners may be small, we complement this figure with the regrets on the right panel of box-plots, i.e., final error rate of the chosen learner minus the minimum of final error rate (in the log-scale).

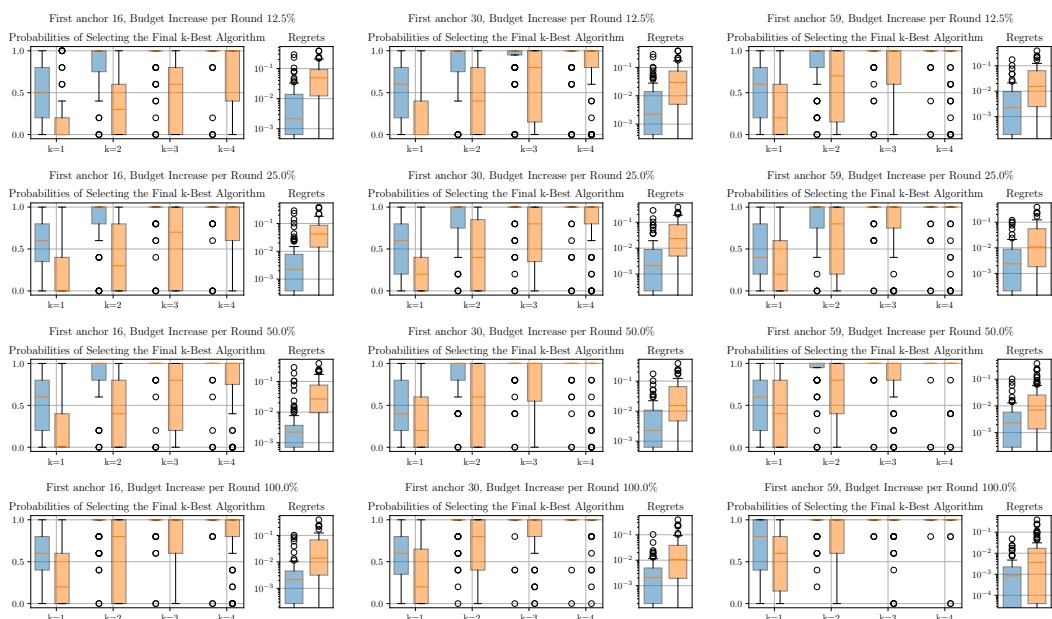

Figure 20: Results of SH for varying starting anchors (with 16, 30, 59 training instances, respectively) with varying budgets (12.5%, 25.0%, 50.0%, and 100%).

The figures nicely confirm the intuition on the relevance of learning curve crossing and the performance of SH. In the group of learners whose curves rarely cross (blue), the algorithm almost always picks the best or at least runner-up. For learners that frequently cross this is less often the case. The regrets show a similar pattern. As such, we can observe that crossing curves make model selection using SH significantly more challenging. The difference between the groups is also nicely reflected in the regrets, which are significantly better in the group of learners whose curves rarely cross compared to the ones where curves frequently cross.

When SH starts at the first available anchor (16 training instances), its ability to identify top-performing algorithms is particularly poor for the frequently-crossing group: in approximately 80% of cases, it fails to select either the best or the second-best candidate. This suggests that critical curve crossings likely occur at very early stages, causing premature elimination of ultimately superior learners.

However, this issue diminishes as the starting anchor increases. When starting at the 16th anchor (59 training instances), the performance gap narrows, and SH becomes more effective even for the frequently-crossing group. These results demonstrate that crossing learning curves pose a serious challenge for multi-fidelity optimization strategies like SH, especially when early budgets dominate the selection process.

# H    Alternative Method To Detect Monotonicity Violations

Monotonicity can also be assessed at the local level by examining trends between consecutive anchors. We introduce a method to statistically identify local monotonicity, classifying all segments between consecutive anchor pairs into three categories: significant improvement, significant worsening, and insignificant change. This method evaluates all consecutive segments of the learning curve and classifies each segment into one of three categories: *improvement*, *worsening*, or *insignificant*, based on statistical significance of paired *t*-test (an example in Figure 21a).

By leveraging the local monotonicity of consecutive anchors, we propose an alternative approach to detect the occurrence of the peaking phenomenon. Specifically, we assume that there is always at least a peaking occurrence between a pair of *improvement* and *worsening* segments, potentially interspersed with *insignificant* status in between. Based on this, we define a criterion for detecting peaks by examining such anchor status transitions across the learning curve. The resulting detection, illustrated in Figure 21b, provides a complementary perspective to the main method. Although this approach does not employ the Bonferroni correction and is therefore slightly more permissive, the estimated probabilities of peak occurrences remain broadly consistent with Figure 4.

However, due to our conservative stance toward identifying ill-behaved learning curve shapes, we opt for a more statistically rigorous approach. We do not adopt the peak detection method by using local monotonicity described above, as it does not incorporate multiple comparison corrections and may be prone to false positives.

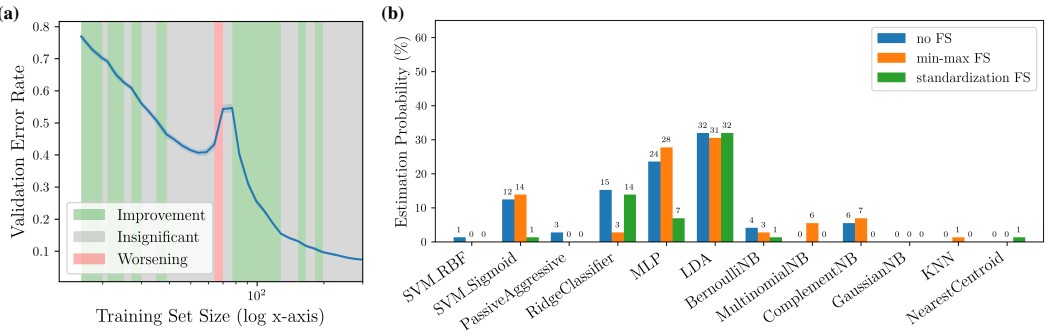

Figure 21: Illustration of (a) local monotonicity identification and (b) the results of using local monotonicity to detect statistic peaking.

# I    Statistical Correction Using Holm's Method

We consider to use a slightly less conservative method called Holm's Step-Down Procedure (Holm's method) [18], in the sense that it will reject more null hypotheses, typically resulting in fewer Type II errors. As shown in Table 9, the results remain consistent with our original findings: an even larger fraction of learning curves are identified as ill-behaved, up to 19%.

Table 9: Ill-behavior statistics of the LCDB 1.1 (23 learners, Dummy excluded) by using Holm's method.

| Shapes / Database | LCDB 1.1 CC-18 (72) | | | LCDB 1.1 FULL (265) | | |
|---|---|---|---|---|---|---|
| | no FS | min-max FS | standardization FS | no FS | min-max FS | standardization FS |
| Missing | 1.9% | 0.0% | 8.7% | 2.6% | 0.0% | 8.7% |
| Non-Monotone ($\neg$ M) | 13.3% | 13.0% | 10.6% | 13.2% | 14.1% | 11.8% |
| Non-Convex ($\neg$ C) | 14.6% | 14.1% | 11.8% | 14.8% | 15.4% | 13.5% |
| Ill-behaved ($\neg$ M $\cup$ $\neg$ C) | 18.2% | 17.3% | 14.7% | 18.8% | 19.8% | 16.8% |

## J  Broader Discussion about LCDB 1.1

**Resource Usage and Green Machine Learning.**  The creation of LCDB 1.1 involved approximately 800,000 CPU-hours and 3,000 GPU-hours of computing. The computing was conducted on a heterogeneous cluster environment provided by the Delft Artificial Intelligence Cluster (DAIC) [77]. Since the specific CPU node used for each job was not fixed, jobs were scheduled across a range of CPU nodes: AMD EPYC 7502P 32-Core Processor, AMD EPYC 9534 64-Core Processor, AMD EPYC 7413 24-Core Processor, AMD EPYC 7452 32-Core Processor, AMD EPYC 7543 32-Core Processor, Intel(R) Xeon(R) CPU E5-2667 v4 @ 3.20GHz, Intel(R) Xeon(R) CPU E5-2680 v4 @ 2.40GHz, Intel(R) Xeon(R) Gold 5218 CPU @ 2.30GHz, Intel(R) Xeon(R) Gold 6130 CPU @ 2.10GHz, Intel(R) Xeon(R) Gold 6140 CPU @ 2.30GHz. For TabPFN v2 learning curves, we used DelftBlue Supercomputer hosted at the Delft High Performance Computing Centre (DHPC) [78], utilizing Nvidia A100 and Nvidia Tesla V100. Each compute job was allocated 20 GB of memory. We set the maximum execution time as 3.5 hours for computing per anchor, inner, and outer seed combination. If the total runtime exceeds this limit, the job is terminated and its status is recorded as a timeout. Both the error message and execution status were logged.

We acknowledge the environmental footprint of this computation and have taken steps to reuse existing model outputs where possible to minimize redundant training. All prediction results during our training process are stored, which allows us to obtain more variants in addition to error rate learning curves, such as the AUC curve, the F1 score learning curve, the log-loss learning curve, and some more types of learning curves in metrics that may be of interest in the future. Furthermore, the processed learning curve data, LCDB 1.1, are annotated with standardized metadata using the Croissant format [79], which facilitates dataset discoverability and machine-readability through integration with web-based data indexing systems.

**Social Impact.**  LCDB 1.1 provides researchers with a perspective from learning curve to better understand the relationship between model performance and the amount of training data. This can be especially valuable in domains where data collection is costly or limited, such as medicine [80], potentially improving outcomes in areas with high social impact. Furthermore, from a meta-learning standpoint, LCDB 1.1 could be a challenging benchmark contributing toward more efficient and automated machine learning systems, which can democratize access to high-quality models in fields that traditionally require significant expert knowledge. However, as with many powerful tools, LCDB 1.1 also poses dual-use concerns. Insights derived from learning curves and dataset performance may inadvertently aid malicious applications such as targeted misinformation or the development of weapons systems.

## K  OpenML Dataset List in LCDB 1.1

Since data curation is far from trivial, even for tabular data [81], we show the following basic properties of the dataset used in LCDB 1.1: OpenML ID (ID), name of the dataset (Name), number of features (#Features), number of samples (#Samples), number of classes (#Classes), and the maximum class ratio (Ratio). Hopefully, this summary can facilitate an assessment of the composition of LCDB 1.1 and allows for the identification of relevant subsets for further analysis.

Table 10: A total overview of 265 OpenML datasets used in LCDB 1.1 FULL.

| ID | Name | #Features | #Samples | #Classes | Ratio |
|---|---|---|---|---|---|
| 3 | kr-vs-kp | 36 | 3196 | 2 | 0.52 |
| 6 | letter | 16 | 20000 | 26 | 0.04 |
| 11 | balance-scale | 4 | 625 | 3 | 0.46 |
| 12 | mfeat-factors | 216 | 2000 | 10 | 0.10 |
| 13 | breast-cancer | 9 | 286 | 2 | 0.70 |
| 14 | mfeat-fourier | 76 | 2000 | 10 | 0.10 |
| 15 | breast-w | 9 | 699 | 2 | 0.66 |
| 16 | mfeat-karhunen | 64 | 2000 | 10 | 0.10 |
| 18 | mfeat-morphological | 6 | 2000 | 10 | 0.10 |
| 21 | car | 6 | 1728 | 4 | 0.70 |
| 22 | mfeat-zernike | 47 | 2000 | 10 | 0.10 |
| 23 | cmc | 9 | 1473 | 3 | 0.43 |
| 24 | mushroom | 22 | 8124 | 2 | 0.52 |
| 26 | nursery | 8 | 12960 | 5 | 0.33 |
| 28 | optdigits | 64 | 5620 | 10 | 0.10 |
| 29 | credit-approval | 15 | 690 | 2 | 0.56 |
| 30 | page-blocks | 10 | 5473 | 5 | 0.90 |
| 31 | credit-g | 20 | 1000 | 2 | 0.70 |
| 32 | pendigits | 16 | 10992 | 10 | 0.10 |
| 36 | segment | 19 | 2310 | 7 | 0.14 |
| 37 | diabetes | 8 | 768 | 2 | 0.65 |
| 38 | sick | 29 | 3772 | 2 | 0.94 |
| 44 | spambase | 57 | 4601 | 2 | 0.61 |
| 46 | splice | 60 | 3190 | 3 | 0.52 |
| 50 | tic-tac-toe | 9 | 958 | 2 | 0.65 |
| 54 | vehicle | 18 | 846 | 4 | 0.26 |
| 55 | hepatitis | 19 | 155 | 2 | 0.79 |
| 57 | hypothyroid | 29 | 3772 | 4 | 0.92 |
| 60 | waveform-5000 | 40 | 5000 | 3 | 0.34 |
| 61 | iris | 4 | 150 | 3 | 0.33 |
| 151 | electricity | 8 | 45312 | 2 | 0.58 |
| 179 | adult | 14 | 48842 | 2 | 0.76 |
| 180 | covertype | 54 | 110393 | 7 | 0.47 |
| 181 | yeast | 8 | 1484 | 10 | 0.31 |
| 182 | satimage | 36 | 6430 | 6 | 0.24 |
| 184 | kropt | 6 | 28056 | 18 | 0.16 |
| 185 | baseball | 16 | 1340 | 3 | 0.91 |
| 188 | eucalyptus | 19 | 736 | 5 | 0.29 |
| 201 | pol | 48 | 15000 | 11 | 0.62 |
| 273 | IMDB.drama | 1001 | 120919 | 2 | 0.64 |
| 293 | covertype | 54 | 581012 | 2 | 0.51 |
| 299 | libras_move | 90 | 360 | 15 | 0.07 |
| 300 | isolet | 617 | 7797 | 26 | 0.04 |
| 307 | vowel | 12 | 990 | 11 | 0.09 |
| 336 | SPECT | 22 | 267 | 2 | 0.79 |
| 346 | aids | 4 | 50 | 2 | 0.50 |
| 351 | codrna | 8 | 488565 | 2 | 0.67 |
| 354 | poker | 10 | 1025010 | 2 | 0.50 |
| 357 | vehicle_sensIT | 100 | 98528 | 2 | 0.50 |
| 380 | SyskillWebert-Bands | 2 | 61 | 3 | 0.64 |
| 389 | fbis.wc | 2000 | 2463 | 17 | 0.21 |
| 390 | new3s.wc | 26832 | 9558 | 44 | 0.07 |
| 391 | re0.wc | 2886 | 1504 | 13 | 0.40 |
| 392 | oh0.wc | 3182 | 1003 | 10 | 0.19 |
| 393 | la2s.wc | 12432 | 3075 | 6 | 0.29 |
| 395 | re1.wc | 3758 | 1657 | 25 | 0.22 |
| 396 | la1s.wc | 13195 | 3204 | 6 | 0.29 |
| 398 | wap.wc | 8460 | 1560 | 20 | 0.22 |
| 399 | ohscal.wc | 11465 | 11162 | 10 | 0.15 |
| 401 | oh10.wc | 3238 | 1050 | 10 | 0.16 |
| 446 | prnn_crabs | 7 | 200 | 2 | 0.50 |
| 458 | analcatdata_authorship | 70 | 841 | 4 | 0.38 |

| 469 | analcatdata_dmft | 4 | 797 | 6 | 0.19 |
|---|---|---|---|---|---|
| 554 | mnist_784 | 784 | 70000 | 10 | 0.11 |
| 679 | rmftsa_sleepdata | 2 | 1024 | 4 | 0.39 |
| 715 | fri_c3_1000_25 | 25 | 1000 | 2 | 0.56 |
| 718 | fri_c4_1000_100 | 100 | 1000 | 2 | 0.56 |
| 720 | abalone | 8 | 4177 | 2 | 0.50 |
| 722 | pol | 48 | 15000 | 2 | 0.66 |
| 723 | fri_c4_1000_25 | 25 | 1000 | 2 | 0.55 |
| 727 | 2dplanes | 10 | 40768 | 2 | 0.50 |
| 728 | analcatdata_supreme | 7 | 4052 | 2 | 0.76 |
| 734 | ailerons | 40 | 13750 | 2 | 0.58 |
| 735 | cpu_small | 12 | 8192 | 2 | 0.70 |
| 737 | space_ga | 6 | 3107 | 2 | 0.50 |
| 740 | fri_c3_1000_10 | 10 | 1000 | 2 | 0.56 |
| 741 | rmftsa_sleepdata | 2 | 1024 | 2 | 0.50 |
| 743 | fri_c1_1000_5 | 5 | 1000 | 2 | 0.54 |
| 751 | fri_c4_1000_10 | 10 | 1000 | 2 | 0.56 |
| 752 | puma32H | 32 | 8192 | 2 | 0.50 |
| 761 | cpu_act | 21 | 8192 | 2 | 0.70 |
| 772 | quake | 3 | 2178 | 2 | 0.56 |
| 797 | fri_c4_1000_50 | 50 | 1000 | 2 | 0.56 |
| 799 | fri_c0_1000_5 | 5 | 1000 | 2 | 0.50 |
| 803 | delta_ailerons | 5 | 7129 | 2 | 0.53 |
| 806 | fri_c3_1000_50 | 50 | 1000 | 2 | 0.56 |
| 807 | kin8nm | 8 | 8192 | 2 | 0.51 |
| 813 | fri_c3_1000_5 | 5 | 1000 | 2 | 0.56 |
| 816 | puma8NH | 8 | 8192 | 2 | 0.50 |
| 819 | delta_elevators | 6 | 9517 | 2 | 0.50 |
| 821 | house_16H | 16 | 22784 | 2 | 0.70 |
| 822 | cal_housing | 8 | 20640 | 2 | 0.59 |
| 823 | houses | 8 | 20640 | 2 | 0.57 |
| 833 | bank32nh | 32 | 8192 | 2 | 0.69 |
| 837 | fri_c1_1000_50 | 50 | 1000 | 2 | 0.55 |
| 843 | house_8L | 8 | 22784 | 2 | 0.70 |
| 845 | fri_c0_1000_10 | 10 | 1000 | 2 | 0.51 |
| 846 | elevators | 18 | 16599 | 2 | 0.69 |
| 847 | wind | 14 | 6574 | 2 | 0.53 |
| 849 | fri_c0_1000_25 | 25 | 1000 | 2 | 0.50 |
| 866 | fri_c2_1000_50 | 50 | 1000 | 2 | 0.58 |
| 871 | pollen | 5 | 3848 | 2 | 0.50 |
| 881 | mv | 10 | 40768 | 2 | 0.60 |
| 897 | colleges_aaup | 15 | 1161 | 2 | 0.70 |
| 901 | fried | 10 | 40768 | 2 | 0.50 |
| 903 | fri_c2_1000_25 | 25 | 1000 | 2 | 0.56 |
| 904 | fri_c0_1000_50 | 50 | 1000 | 2 | 0.51 |
| 910 | fri_c1_1000_10 | 10 | 1000 | 2 | 0.56 |
| 912 | fri_c2_1000_5 | 5 | 1000 | 2 | 0.58 |
| 913 | fri_c2_1000_10 | 10 | 1000 | 2 | 0.58 |
| 914 | balloon | 1 | 2001 | 2 | 0.76 |
| 917 | fri_c1_1000_25 | 25 | 1000 | 2 | 0.55 |
| 923 | visualizing_soil | 4 | 8641 | 2 | 0.55 |
| 930 | colleges_usnews | 33 | 1302 | 2 | 0.53 |
| 934 | socmob | 5 | 1156 | 2 | 0.78 |
| 953 | splice | 60 | 3190 | 2 | 0.52 |
| 958 | segment | 19 | 2310 | 2 | 0.86 |
| 959 | nursery | 8 | 12960 | 2 | 0.67 |
| 962 | mfeat-morphological | 6 | 2000 | 2 | 0.90 |
| 966 | analcatdata_halloffame | 16 | 1340 | 2 | 0.91 |
| 971 | mfeat-fourier | 76 | 2000 | 2 | 0.90 |
| 976 | JapaneseVowels | 14 | 9961 | 2 | 0.84 |
| 977 | letter | 16 | 20000 | 2 | 0.96 |
| 978 | mfeat-factors | 216 | 2000 | 2 | 0.90 |
| 979 | waveform-5000 | 40 | 5000 | 2 | 0.66 |
| 980 | optdigits | 64 | 5620 | 2 | 0.90 |
| 991 | car | 6 | 1728 | 2 | 0.70 |

| 993 | kdd_ipums_la_97-small | 60 | 7019 | 2 | 0.63 |
|---|---|---|---|---|---|
| 995 | mfeat-zernike | 47 | 2000 | 2 | 0.90 |
| 1000 | hypothyroid | 29 | 3772 | 2 | 0.92 |
| 1002 | ipums_la_98-small | 55 | 7485 | 2 | 0.89 |
| 1018 | ipums_la_99-small | 56 | 8844 | 2 | 0.94 |
| 1019 | pendigits | 16 | 10992 | 2 | 0.90 |
| 1020 | mfeat-karhunen | 64 | 2000 | 2 | 0.90 |
| 1021 | page-blocks | 10 | 5473 | 2 | 0.90 |
| 1036 | sylva_agnostic | 216 | 14395 | 2 | 0.94 |
| 1040 | sylva_prior | 108 | 14395 | 2 | 0.94 |
| 1041 | gina_prior2 | 784 | 3468 | 10 | 0.11 |
| 1042 | gina_prior | 784 | 3468 | 2 | 0.51 |
| 1049 | pc4 | 37 | 1458 | 2 | 0.88 |
| 1050 | pc3 | 37 | 1563 | 2 | 0.90 |
| 1053 | jm1 | 21 | 10885 | 2 | 0.81 |
| 1056 | mc1 | 38 | 9466 | 2 | 0.99 |
| 1063 | kc2 | 21 | 522 | 2 | 0.80 |
| 1067 | kc1 | 21 | 2109 | 2 | 0.85 |
| 1068 | pc1 | 21 | 1109 | 2 | 0.93 |
| 1069 | pc2 | 36 | 5589 | 2 | 1.00 |
| 1083 | mouseType | 45101 | 214 | 7 | 0.32 |
| 1084 | BurkittLymphoma | 22283 | 220 | 3 | 0.58 |
| 1085 | anthracyclineTaxaneChemotherapy | 61359 | 159 | 2 | 0.60 |
| 1086 | ovarianTumour | 54621 | 283 | 3 | 0.87 |
| 1087 | hepatitisC | 54621 | 283 | 3 | 0.87 |
| 1088 | variousCancers_final | 54675 | 383 | 10 | 0.40 |
| 1116 | musk | 167 | 6598 | 2 | 0.85 |
| 1119 | adult-census | 14 | 32561 | 2 | 0.76 |
| 1120 | MagicTelescope | 10 | 19020 | 2 | 0.65 |
| 1128 | OVA_Breast | 10935 | 1545 | 2 | 0.78 |
| 1130 | OVA_Lung | 10935 | 1545 | 2 | 0.92 |
| 1134 | OVA_Kidney | 10935 | 1545 | 2 | 0.83 |
| 1138 | OVA_Uterus | 10935 | 1545 | 2 | 0.92 |
| 1139 | OVA_Omentum | 10935 | 1545 | 2 | 0.95 |
| 1142 | OVA_Endometrium | 10935 | 1545 | 2 | 0.96 |
| 1146 | OVA_Prostate | 10935 | 1545 | 2 | 0.96 |
| 1161 | OVA_Colon | 10935 | 1545 | 2 | 0.81 |
| 1166 | OVA_Ovary | 10935 | 1545 | 2 | 0.87 |
| 1216 | Click_prediction_small | 9 | 1496391 | 2 | 0.96 |
| 1233 | eating | 6373 | 945 | 7 | 0.15 |
| 1235 | Agrawal1 | 9 | 1000000 | 2 | 0.67 |
| 1236 | Stagger1 | 3 | 1000000 | 2 | 0.89 |
| 1441 | KungChi3 | 39 | 123 | 2 | 0.87 |
| 1448 | KnuggetChase3 | 39 | 194 | 2 | 0.81 |
| 1450 | MindCave2 | 39 | 125 | 2 | 0.65 |
| 1457 | amazon-commerce-reviews | 10000 | 1500 | 50 | 0.02 |
| 1461 | bank-marketing | 16 | 45211 | 2 | 0.88 |
| 1462 | banknote-authentication | 4 | 1372 | 2 | 0.56 |
| 1464 | blood-transfusion-service-center | 4 | 748 | 2 | 0.76 |
| 1465 | breast-tissue | 9 | 106 | 6 | 0.21 |
| 1468 | cnae-9 | 856 | 1080 | 9 | 0.11 |
| 1475 | first-order-theorem-proving | 51 | 6118 | 6 | 0.42 |
| 1477 | gas-drift-different-concentrations | 129 | 13910 | 6 | 0.22 |
| 1478 | har | 561 | 10299 | 6 | 0.19 |
| 1479 | hill-valley | 100 | 1212 | 2 | 0.50 |
| 1480 | ilpd | 10 | 583 | 2 | 0.71 |
| 1483 | ldpa | 7 | 164860 | 11 | 0.33 |
| 1485 | madelon | 500 | 2600 | 2 | 0.50 |
| 1486 | nomao | 118 | 34465 | 2 | 0.71 |
| 1487 | ozone-level-8hr | 72 | 2534 | 2 | 0.94 |
| 1488 | parkinsons | 22 | 195 | 2 | 0.75 |
| 1489 | phoneme | 5 | 5404 | 2 | 0.71 |
| 1494 | qsar-biodeg | 41 | 1055 | 2 | 0.66 |
| 1497 | wall-robot-navigation | 24 | 5456 | 4 | 0.40 |
| 1499 | seeds | 7 | 210 | 3 | 0.33 |

| 1501 | semeion | 256 | 1593 | 10 | 0.10 |
|---|---|---|---|---|---|
| 1503 | spoken-arabic-digit | 14 | 263256 | 10 | 0.10 |
| 1509 | walking-activity | 4 | 149332 | 22 | 0.15 |
| 1510 | wdbc | 30 | 569 | 2 | 0.63 |
| 1515 | micro-mass | 1300 | 571 | 20 | 0.11 |
| 1566 | hill-valley | 100 | 1212 | 2 | 0.50 |
| 1567 | poker-hand | 10 | 1025009 | 10 | 0.50 |
| 1575 | ijcnn | 22 | 191681 | 2 | 0.90 |
| 1590 | adult | 14 | 48842 | 2 | 0.76 |
| 1592 | aloi | 128 | 108000 | 1000 | 0.00 |
| 1597 | creditcard | 29 | 284807 | 2 | 1.00 |
| 4134 | Bioresponse | 1776 | 3751 | 2 | 0.54 |
| 4135 | Amazon_employee_access | 9 | 32769 | 2 | 0.94 |
| 4137 | Dorothea | 100000 | 1150 | 2 | 0.90 |
| 4534 | PhishingWebsites | 30 | 11055 | 2 | 0.56 |
| 4538 | GesturePhaseSegmentationProcessed | 32 | 9873 | 5 | 0.30 |
| 4541 | Diabetes130US | 49 | 101766 | 3 | 0.54 |
| 6332 | cylinder-bands | 37 | 540 | 2 | 0.58 |
| 23381 | dresses-sales | 12 | 500 | 2 | 0.58 |
| 23512 | higgs | 28 | 98050 | 2 | 0.53 |
| 23517 | numerai28.6 | 21 | 96320 | 2 | 0.51 |
| 40498 | wine-quality-white | 11 | 4898 | 7 | 0.45 |
| 40499 | texture | 40 | 5500 | 11 | 0.09 |
| 40664 | car-evaluation | 21 | 1728 | 4 | 0.70 |
| 40668 | connect-4 | 42 | 67557 | 3 | 0.66 |
| 40670 | dna | 180 | 3186 | 3 | 0.52 |
| 40672 | fars | 29 | 100968 | 8 | 0.42 |
| 40677 | led24 | 24 | 3200 | 10 | 0.11 |
| 40685 | shuttle | 9 | 58000 | 7 | 0.79 |
| 40687 | solar-flare | 12 | 1066 | 6 | 0.31 |
| 40701 | churn | 20 | 5000 | 2 | 0.86 |
| 40713 | dis | 29 | 3772 | 2 | 0.98 |
| 40900 | Satellite | 36 | 5100 | 2 | 0.99 |
| 40910 | Speech | 400 | 3686 | 2 | 0.98 |
| 40923 | Devnagari-Script | 1024 | 92000 | 46 | 0.02 |
| 40927 | CIFAR_10 | 3072 | 60000 | 10 | 0.10 |
| 40966 | MiceProtein | 77 | 1080 | 8 | 0.14 |
| 40971 | collins | 19 | 1000 | 30 | 0.08 |
| 40975 | car | 6 | 1728 | 4 | 0.70 |
| 40978 | Internet-Advertisements | 1558 | 3279 | 2 | 0.86 |
| 40979 | mfeat-pixel | 240 | 2000 | 10 | 0.10 |
| 40981 | Australian | 14 | 690 | 2 | 0.56 |
| 40982 | steel-plates-fault | 27 | 1941 | 7 | 0.35 |
| 40983 | wilt | 5 | 4839 | 2 | 0.95 |
| 40984 | segment | 16 | 2310 | 7 | 0.14 |
| 40994 | climate-model-simulation-crashes | 18 | 540 | 2 | 0.91 |
| 40996 | Fashion-MNIST | 784 | 70000 | 10 | 0.10 |
| 41027 | jungle_chess_2pcs_raw_endgame_complete | 6 | 44819 | 3 | 0.51 |
| 41142 | christine | 1636 | 5418 | 2 | 0.50 |
| 41143 | jasmine | 144 | 2984 | 2 | 0.50 |
| 41144 | madeline | 259 | 3140 | 2 | 0.50 |
| 41145 | philippine | 308 | 5832 | 2 | 0.50 |
| 41146 | sylvine | 20 | 5124 | 2 | 0.50 |
| 41150 | MiniBooNE | 50 | 130064 | 2 | 0.72 |
| 41156 | ada | 48 | 4147 | 2 | 0.75 |
| 41157 | arcene | 10000 | 100 | 2 | 0.56 |
| 41158 | gina | 970 | 3153 | 2 | 0.51 |
| 41159 | guillermo | 4296 | 20000 | 2 | 0.60 |
| 41161 | riccardo | 4296 | 20000 | 2 | 0.75 |
| 41163 | dilbert | 2000 | 10000 | 5 | 0.20 |
| 41164 | fabert | 800 | 8237 | 7 | 0.23 |
| 41165 | robert | 7200 | 10000 | 10 | 0.10 |
| 41166 | volkert | 180 | 58310 | 10 | 0.22 |
| 41167 | dionis | 60 | 416188 | 355 | 0.01 |
| 41168 | jannis | 54 | 83733 | 4 | 0.46 |

| 41169 | helena | 27 | 65196 | 100 | 0.06 |
|---|---|---|---|---|---|
| 41228 | Klaverjas2018 | 32 | 981541 | 2 | 0.54 |
| 41972 | Indian_pines | 220 | 9144 | 8 | 0.44 |
| 42734 | okcupid-stem | 19 | 50789 | 3 | 0.72 |
| 42742 | porto-seguro | 57 | 595212 | 2 | 0.96 |
| 42769 | Higgs | 28 | 1000000 | 2 | 0.53 |
| 42809 | kits | 27648 | 1000 | 2 | 0.52 |
| 42810 | PCam | 27648 | 4000 | 2 | 0.51 |

