# LCDB 1.1: A Database Illustrating Learning Curves Are More Ill-Behaved Than Previously Thought

**Cheng Yan**[1]    **Felix Mohr**[2]    **Tom Viering**[1]
[1]Delft University of Technology    [2]Universidad de La Sabana
{c.yan-1, t.j.viering}@tudelft.nl   felix.mohr@unisabana.edu.co

## 1 A Note Regarding Dataset Hosting

For the actual appendices, please see the main paper submission. Here, we would like to make a few notes regarding the dataset hosting.

**Self-Hosting Platform**   Our dataset is self-hosted on the 4TU.ResearchData platform, a trusted institutional repository based in the Netherlands, which guarantees long-term preservation of research data for a minimum of 15 years.[1]

**Data Access Note**   We provide a public access link (also attached in the main submission).[2]

**Machine Access via Croissant Metadata**   For machine access, Croissant metadata file can be found in our GitHub repository.[3]

---

[1]https://data.4tu.nl/
[2]https://doi.org/10.4121/3bd18108-fad0-4e4c-affd-4341fba99306
[3]https://github.com/learning-curve-research/LCDB-1.1/blob/main/croissant.json