# OpenReview forum: "LCDB 1.1: A Database Illustrating Learning Curves Are More Ill-Behaved Than Previously Thought"
_NeurIPS.cc/2025/Datasets_and_Benchmarks_Track — NeurIPS 2025 Datasets and Benchmarks Track poster_

### Official Review · Reviewer_KUV1 · 2025-06-24

**Rating:** 4
**Confidence:** 1

**Summary:**

This study introduces LCDB 1.1, a large-scale database of high-resolution learning curves, revealing that around 14% exhibit significant irregularities—nearly double previous estimates. The findings show that some learners are more prone to ill-behaved curves and that feature scaling rarely resolves the issue. These irregularities hinder tasks like model selection and curve fitting, making LCDB 1.1 a valuable benchmark for future research.

**Dataset Code Accessibility:**

Yes

**Ethical Considerations:**

No, there are no or only very minor ethics concerns

**Limitations Weaknesses:**

1. The root causes of ill-behaved learning curves remain largely unexplained, making this an open problem for future work.

2. The study does not investigate whether ill-behavior persists under hyperparameter optimization, which limits the conclusions—though the authors justify this due to computational cost.

**Strengths Contributions:**

1. The problem discussed in the paper is of great importance for studying the interpretability of the optimization process of the model.

2. LCDB 1.1 improves upon LCDB 1.0 by addressing the issue of overly easy datasets that led to flat, non-informative learning curves. It incorporates the OpenML-CC18 benchmark, which offers a more challenging and carefully curated set of datasets. LCDB 1.1 FULL combines these with all datasets from LCDB 1.0. Additionally, the number of learners has been expanded to 28, including mixed Naive Bayes variants to handle one-hot features and a dummy classifier as a weak baseline, resulting in a more diverse and robust evaluation framework.

3. The study uses rigorous statistical techniques (e.g., Bonferroni correction) to identify monotonicity and convexity violations in learning curves, though acknowledging the trade-offs. The observed ill-behaviors are shown to have real implications for downstream tasks like model selection and learning curve fitting, emphasizing the practical utility of LCDB 1.1.

---

> ### Author Rebuttal · Authors · 2025-07-30
>
> We thank the reviewer for emphasizing the importance of our work in studying interpretability and appreciating the improvements introduced in LCDB 1.1.
>
> While the root causes of ill-behaved curves remain an open question, we now provide further discussion.
>
> **The root causes of ill-behaved learning curves remain largely unexplained, making this an open problem for future work.**
>
>
> We agree this would strengthen the paper. However, it should be clear that this is a non-trivial ask. We would like to point out that, that such theoretical analysis often needs to be carried out per learner, and it is not clear how that should be carried out for all learners. Only alone for the peaking phenomema, various papers have studied this from different viewpoints, see for example [1-11]. This paper [12] gives an overview of all known theory papers regarding ill-behavior of learning curves. Several papers have also already been written about how to mitigate such behaviors (see [12]), but all these works are still mostly in the realm of learning theory, far from practice.
>
> By openly publishing the LCDB 1.1, and underscoring that ill-behavior is more prominent than previously thought, we want to bring these learning phemomema under the attention of the broader machine learning community. We believe this will lead to more researchers looking into the various ill-behaviors observed, which will hopefully lead to a better understanding of these ill-behaviors and how to avoid and mitigate them.
>
> **The study does not investigate whether ill-behavior persists under hyperparameter optimization, which limits the conclusions—though the authors justify this due to computational cost.**
>
> While large-scale hyperparameter tuning is required to fully answer this question, we provide a simple illustrative example: the Nearest Centroid classifier, which has virtually no tunable hyperparameters, yet consistently exhibits ill-behaved learning curves. This suggests that some ill-behaviors may be inherent to the learner’s inductive bias and cannot be easily addressed through hyperparameter tuning.
>
> **References**
>
> [1] F. Vallet et al., “Linear and nonlinear extension of the pseudoinverse solution for learning boolean functions,” Europhys. Lett., vol. 9, no. 4, p. 315, 1989.
>
> [2] Duin, R. P. (2000, September). Classifiers in almost empty spaces. In Proceedings 15th International Conference on Pattern Recognition. ICPR-2000 (Vol. 2, pp. 1-7). IEEE.
>
> [3] Zollanvari, A., James, A. P., & Sameni, R. (2020). A theoretical analysis of the peaking phenomenon in classification. Journal of Classification, 37(2), 421-434.
>
> [4] Belkin, M., Hsu, D., Ma, S., & Mandal, S. (2019). Reconciling modern machine-learning practice and the classical bias–variance trade-off. Proceedings of the National Academy of Sciences, 116(32), 15849-15854.
>
> [5] Opper, M., Kinzel, W., Kleinz, J., & Nehl, R. (1990). On the ability of the optimal perceptron to generalise. Journal of Physics A: Mathematical and General, 23(11), L581.
>
> [6] T. L. Watkin et al., “The statistical mechanics of learning a rule,”Rev. Mod. Phys., vol. 65, no. 2, p. 499, 1993.
>
> [7] A. Engel and C. Van den Broeck, Statistical mechanics of learning. Cambridge University Press, 2001.
>
> [8] S. Raudys and R. Duin, “Expected classification error of the Fisher linear classifier with pseudo-inverse covariance matrix,”Pattern Recognit. Lett., vol. 19, no. 5-6, pp. 385–392, 1998.
>
> [9] P. Nakkiran et al., “Optimal regularization can mitigate double descent,” arXiv:2003.01897, 2020.
>
> [10] M. S. Advani and A. M. Saxe, “High-dimensional dynamics of generalization error in neural networks,” arXiv:1710.03667, 2017.
>
> [11] T. Hastie et al., “Surprises in high-dimensional ridgeless least squares interpolation,” arXiv:1903.08560, 2019.
>
> [12] Viering, T., & Loog, M. (2022). The shape of learning curves: a review. IEEE Transactions on Pattern Analysis and Machine Intelligence, 45(6), 7799-7819.

---

> > ### Comment · Reviewer_KUV1 · 2025-08-06
> > **Response to rebuttal**
> >
> > Thank you for your reply. Your reply answered my questions. I still keep my score and confidence.

---

### Official Review · Reviewer_opAM · 2025-06-30

**Rating:** 5
**Confidence:** 5

**Summary:**

This paper introduces LCDB 1.1, an enhanced database of sample-wise learning curves in classification. The authors rigorously analyze curve behaviors and find that about 14% of curves are statistically significant ill-behavior (non-monotonicity or non-convexity), nearly double prior estimates, and identify high-ill-behavior learners (e.g., Sigmoid SVM, LDA, QDA). What's more, feature scaling rarely mitigates ill-behavior and sometimes worsens it.

**Dataset Code Accessibility:**

Yes

**Ethical Considerations:**

No, there are no or only very minor ethics concerns

**Limitations Weaknesses:**

- While the focus on classification is justified, the paper would benefit from a brief discussion on how the findings might or might not generalize to regression or other supervised learning settings.
- The interpretation of “peaking,” “dipping,” and “phase transitions” need formal definitions. Some of these terms may benefit from additional visual or algorithmic refinement for more interpretable labeling.

**Strengths Contributions:**

- This paper is significant and novel, since it addresses the critical limitations of LCDB 1.0, and they introduce novel statistical analysis methods to robustly quantify monotonicity and convexity violations.
- The analysis showing that ill-behaved curves negatively impact downstream tasks. I think the community will be interested about the theoretical insight of this phenomenon.
- The presentation is good. I enjoy reading it.

---

> ### Author Rebuttal · Authors · 2025-07-30
>
> We thank the reviewer for highlighting the novelty, theoretical insight, and clarity of our work.
>
> We now include a discussion on generalization to other tasks (such as regression) and have clarified our definitions of “peaking,” and “dipping” formally.
>
> **The paper would benefit from a brief discussion on how the findings might or might not generalize to regression or other supervised learning settings.**
>
> A good point. While LCDB 1.1 is centered on classification, many of the observed learning curve irregularities, such as non-monotonicity, may also arise in regression problems and even unsupervised learning such as clustering. For instance, in Gaussian Process regression under squared loss, learning curves have been shown to exhibit peaking depending on model misspecification [1]. Similar issues have been observed in unsupervised learning as well; notably, even simple methods like k-means can display non-monotonic behavior with more data [2], challenging the assumption that more data always helps.
>
> **The interpretation of “peaking,” “dipping,” and “phase transitions” need formal definitions.**
>
> We agree with the reviewer. Due to lack of space we will add the formal definitions to the Appendix, but we also discuss them here.
>
> Given a learning curve $C(n_i)$ with $N$ points, where $C$ is the loss (lower means better) and where $1 \leq i \leq N$. We define the following phenomena:
>
> Definition (Peaking Phenomenon):
> *Peaking* occurs if there exists a triplet of indices, $1 \leq h < i < j \leq N$, such that:
> $$
> C(n_i) > C(n_h) \quad \text{and} \quad C(n_i) > C(n_j).
> $$
> In this case, $C(n_i)$ forms a local peak, indicating that the model's performance temporarily degrades and subsequently recovers as more data is added.
>
>
> Definition (Dipping Phenomenon):
> *Dipping* occurs if there exists an index $i$, $1 \leq i < N$, such that:
> $$
> C(n_i) < C(n_N) .
> $$
> This indicates a sustained degradation of model performance, with no recovery observed as more data is added.
>
> Note that, for *phase transitions*, there is no clear formal definition that we are aware off. In fact, detecting a phase transition is rather subjective. A phase transition was defined by [3] as: "particular learning curve properties change relatively abruptly", which is already subjective. Often the learning curve has a platea, after which it abruptly improves. This is often characterized by the fact that the curve is non-convex. Otherwise, it is not clear how to formally define this.
>
> **References**
>
> [1] Sollich, P. (2001). Gaussian process regression with mismatched models. Advances in Neural Information Processing Systems, 14.
>
> [2] Loog, M., Krijthe, J. H., & Bicego, M. (2023). Also for k-means: more data does not imply better performance. Machine Learning, 112(8), 3033-3050.
>
> [3] Viering, T., & Loog, M. (2022). The shape of learning curves: a review. IEEE Transactions on Pattern Analysis and Machine Intelligence, 45(6), 7799-7819.

---

> > ### Comment · Area_Chair_6VMM · 2025-08-04
> > **Please engage in discussion**
> >
> > Dear Reviewer,
> > the authors provided a detailed reply to address your feedbacks.
> > It is important to acknowledge their response and engage in a discussion in case you may need of further clarifications to reach your final assessment.
> > Many thanks for your help.
> > Best,
> >
> > AC

---

> > ### Comment · Reviewer_opAM · 2025-08-07
> > **Reply**
> >
> > Thanks for the rebuttal. My concerns are generally addressed and I keep my initial rating.

---

### Official Review · Reviewer_n6w6 · 2025-07-02

**Rating:** 5
**Confidence:** 4

**Summary:**

This paper presents LCDB 1.1, a valuable large-scale, high-resolution database of learning curves from 28 classical learners across 265 tabular datasets. As a substantial upgrade to its predecessor (LCDB 1.0), it features four times the anchor resolution, addresses data leakage, and significantly reduces missing results. Using new, statistically rigorous tests, the paper provides compelling evidence that a significant fraction (~14%) of learning curves are "ill-behaved" (non-monotone or non-convex), a figure nearly double previous estimates. The authors clearly demonstrate the practical consequences of this ill-behavior on downstream tasks like parametric curve fitting and model selection, establishing LCDB 1.1 as a valuable and challenging new benchmark.

**Additional Feedback:**

Given its significant methodological improvements, impactful findings, and high standard of reproducibility, this is a strong and valuable contribution.
My suggestions are focused on improving the long-term impact and usability of this excellent resource.

**Suggestions for Improvement:**
- Please consider releasing the dataset under a more permissive license (e.g., CC BY 4.0, which the source data uses) or offering a dual license to allow for commercial research use. This would greatly expand your potential user base.
- I strongly suggest providing a lightweight CSV or JSON manifest file that summarizes the contents of the database (e.g., lists curves, learners, datasets, and key metadata). This would make the resource much easier to explore.
- In the paper, please add a discussion on the potential for dataset selection bias from OpenML and how that might limit the conclusions of your analysis.

**Questions for the Authors**
- Could you elaborate on the choice of the CC BY-NC-SA 4.0 license, especially given the stated relevance to industrial applications? Was a more permissive license considered?

- In your discussion, you note that the Bonferroni correction is "quite pessimistic". Did you consider or perform an analysis using a less stringent correction method, such as controlling for the False Discovery Rate (FDR), and if so, did it meaningfully change the 14% ill-behavior estimate?

**Dataset Code Accessibility:**

Yes

**Dataset Code Comments:**

- The dataset and code are accessible. The authors provide a link to a repository containing the dataset files, and the analysis code is available on GitHub with a Docker container for reproducibility. The link to the data repository works and the files are downloadable.
- The only minor friction point is that the repository is a "privately published item," which may add a small step for users compared to fully public platforms like Hugging Face or Zenodo, but it does not prevent access.
- The provided Croissant metadata file could also be improved by fully exposing the internal schema of the HDF5 files to better support automated discovery.
- Overall, the authors have met the accessibility requirements for the track.

**Ethical Comments:**

- No major ethical concerns that would prevent publication were identified. The paper itself processes no personally identifiable or sensitive data.
- The authors include a thoughtful "Social Impact" section in Appendix I, discussing potential positive impacts (e.g., in data-scarce domains like medicine) and acknowledging potential dual-use concerns.

**Ethical Considerations:**

No, there are no or only very minor ethics concerns

**Final Justification:**

I have raised my confidence score on the accept standpoint. The authors have thoroughly addressed all the concerns raised in my initial review.

The authors' commitment to improving their work is evident through several key actions:

- Improved Licensing: They changed the license to the community-standard and more accessible CC BY 4.0.

- Strengthened Claims: They conducted additional statistical analysis using Holm's method in response to feedback, further solidifying the paper's empirical claims.

- Expanded Scope: They proactively ran new experiments with modern learners (e.g., TabNet), significantly broadening the scope and relevance of their work.

- Enhanced Usability: They committed to concrete improvements for the dataset's usability, such as adding a manifest file.

Furthermore, their thoughtful response regarding the challenges of fairness and bias in datasets demonstrated a deep engagement with the responsible AI aspects of their work.

The authors' constructive engagement have resolved my initial concerns. I am happy to support for acceptance.

**Limitations Weaknesses:**

The work is very strong, and the remaining weaknesses are minor points of friction that could be addressed to broaden the dataset's impact.

- **Restrictive Licensing:** The dataset is released under a CC BY-NC-SA 4.0 license. While the authors mention the importance of their work for industrial applications, the "Non-Commercial" (NC) clause of the license creates a direct conflict and will likely prevent adoption by industry research labs, thus limiting the dataset's potential impact.
- **Data Format and Usability:** The data is provided exclusively in HDF5 format. While efficient for storage, this presents a usability challenge for researchers who may wish to quickly browse the dataset or preview its structure without specialized tools. Providing a simple, human-readable manifest (e.g., a CSV file summarizing the curves) would be a valuable addition, significantly lowering the barrier to entry for exploration and increasing the dataset's overall ease of use.
- **Gaps in Responsible AI Documentation:** The paper lacks a detailed discussion on potential sources of bias in the underlying OpenML datasets. While the work itself doesn't process sensitive information, the selection of datasets could introduce biases that affect the generality of the conclusions about which algorithms are "ill-behaved." A more thorough datasheet or discussion in the paper would be beneficial.
- **Limited Scope of Learners:** The benchmark focuses exclusively on classical, non-deep learning algorithms. While the authors correctly state that these models are still competitive on tabular data, the exclusion of modern tabular deep learning models (e.g., TabNet, FT-Transformer) narrows the paper's appeal to the broader AutoML research community.

**Strengths Contributions:**

The primary contribution of this work is a significantly improved and valuable dataset asset for the machine learning community, particularly for those studying AutoML, meta-learning, and scaling laws.

**Key strengths:**
- **Significance and Relation to Prior Work:** The submission is a substantial improvement over the prior LCDB 1.0. The authors clearly justify the need for a new version by pointing out limitations in the predecessor, such as lower resolution, missing feature scaling, and data leakage. Table 2 effectively summarizes the impressive upgrades: quadrupled anchor resolution, the addition of feature scaling variants, and a reduction in missing results from 12% to 4%.
- **Methodological Rigor and Novelty:** The paper introduces a more rigorous method for analyzing learning curve shapes. The authors rightly criticize prior work for only comparing consecutive anchors and ignoring anchor scaling. Their new approach, detailed in Section 4, compares all pairs (for monotonicity) or triplets (for convexity) of anchors and uses Bonferroni correction to ensure statistical significance, lending strong credibility to their findings.
- **Impactful and Well-Supported Findings:** The paper convincingly demonstrates that learning curves are more ill-behaved than previously thought. The finding that 14.3% of curves are significantly ill-behaved (non-monotone or non-convex) is a key takeaway. Furthermore, the paper provides excellent analysis demonstrating the downstream impact of this ill-behavior on practical tasks. It shows not only that ill-behaved curves harm performance—significantly increasing MSE for parametric fitting (Figure 5) and making Successive Halving more challenging (Figure 6)—but also why, by linking the failure of common parametric assumptions to these curve shapes.

---

> ### Author Rebuttal · Authors · 2025-07-30
>
> We appreciate the reviewer’s recognition of our dataset’s value, methodological rigor, and impactful findings.
>
> In response to the thoughtful suggestions, we have added results for modern tabular deep learning models and plan to address data usability by including a human-readable CSV manifest and more responsible documentation.
>
> ### 1. Restrictive Licensing
> We agree, and will change the license to CC BY 4.0.
>
> ### 2. Data Format and Usability
> **Providing a simple, human-readable manifest would be a valuable addition.**
>
> We agree, and we will add .csv files for increasing the overall ease of use.
>
> **A lightweight CSV or JSON manifest file that summarizes the contents of the database would make the resource much easier to explore.**
>
> Currently, this is provided only in the code. We agree with the reviewer that this is not convenient. We will add a CSV file including all learners, datasets, and other important meta-data, such as the number of classes in a dataset, the dimensionality, etc. However, we cannot provide this during the peer review process (not allowed).
>
> **The only minor friction point is that the repository is a "privately published item," ... but it does not prevent access.**
>
> We have set this to a "privately published item", because we anticipated that we would make some changes during the review process. After acceptance, we will release this dataset publicly, and then the dataset will be accessible to everyone via its URL (without the navigation procedure that is required now).
>
> **The provided Croissant metadata file could also be improved by fully exposing the internal schema of the HDF5 files to better support automated discovery.**
>
> Currently, there is no support for hdf5 files in Croissant. When this functionality is added to the Croissant format, we will add this. Since CSV files are supported by Croissant, we will include their metadata in the Croissaint files.
>
> ### 3. Gaps in Responsible AI Documentation
> **Please add a discussion on the potential for dataset selection bias from OpenML and how that might limit the conclusions of your analysis.**
>
> This is a very good question, which is very tough to answer --- it is far from trivial to understand the biases in OpenML. Such biases can occur at various levels; which datasets are submitted to OpenML? Which of these did it make into our work? And finally, which biases are in these datasets themselves? Data curation is also far from trivial, even for tabular data [1].
>
> We note that there is still not even a clear consensus on the definition of a tabular dataset ([1] does make a constructive proposal). As such, we think it is best that we will keep tracking the developments of new benchmark suites, and release LCDB 1.1 subsets for new and relevant ones. This is why we have created the OpenML CC-18 subset of the LCDB 1.1 --- these datasets were much more carefully curated than the selection of datasets of the LCDB 1.0.
>
> Furthermore, to enable readers to make up their own mind, we will be sure to include, both in the repository and in the paper (appendix), a table that lists all datasets used and some of their basic properties. This way, the reader can have a clear overview of the datasets used in the creation of the LCDB 1.1, and the user can potentially also easily find the relevant subsets of the LCDB 1.1 that is of their interest.
>
> ### 4. Limited Scope of Learners
>
> we have extended LCDB 1.1 by incorporating a broader set of modern tabular learners: the boosting model CatBoost [2] (2018), deep learning models such as TabNet [3] (2021) and RealMLP [4] (2024), and the foundation model TabPFN v2[5] (2025). According to a very recent study TabArena[6], CatBoost remains a strong SOTA model by default, while RealMLP achieves SOTA performance after tuning and ensembling. TabNet is widely used as a deep learning baseline, and TabPFN v2 is a well-performing foundation model for tabular data. We will update the database (not allowed during peer review). However, to answer your question, we have already computed the statistics regarding ill-behavior, see below.
>
> | Ratio in CC18 (72 datasets) with no feature scaling | Non-Monotone ($\neg$M) | Non-Convex ($\neg$C) | Ill-behaved ($\neg$M $\cup$ $\neg$C)|Peaking|Dipping|
> |-|-|-|-|-|-|
> |CatBoost|0.0%|0.0%|0.0%|0.0%|0.0%|
> |RealMLP|2.8%|1.4%|4.2%|0.0%|0.0%|
> |TabNet|12.5%|65.3%|66.7%|19.4%|5.6%|
> |TabPFN v2 (63 datasets)|0.0%|0.0%|0.0%|0.0%|0.0%|
>
> Note that, for TabPFN v2, it is only evaluated on a subset (63 datasets) of CC18 (72), because it only supports datasets with up to 10k training samples, 500 features, and 10 classes for classification task. For this rebuttal we only report results for CC18 with no feature scaling, but after the rebuttal phase we will generate all results for LCDB 1.1 full. Note that, TabPFN v2 and Real MLP used some of the datasets in CC18 during model development (e.g. tuning hyperparameters).
>
> In the results of our manuscript (see Table 3), we found that ensemble methods and tree-like learners have relatively little ill-behavior. Following this, we expected that CatBoost would show similar statistics as the Gradient Boosting learner of Scikit-learn, and indeed we see that it has very little ill-behavior as well. For the deep learner Real MLP, we find limited ill-behavior; we suspect that the substantial tuning of the hyperparameters that was part of Real MLP's design has alleviated a lot of the ill-behavior that remains present in our MLP baseline (which had 27% ill-behaving curves).
>
> We are not aware of any other works that generate learning curves for tabular foundation models, and as such, we were not very certain what to expect. TabPFN v2, according to the paper's title, was designed for small datasets, and is a Bayesian model which can alleviate overfitting. As such, it may be well-suited to learning curve analysis. This agrees with our findings of very little ill-behavior. In contrast, TabNet shows substantial ill-behavior, including various phase transitions. Clearly, this model was developed for larger datasets, because for larger sample sizes, often the learning curve improves significantly. It suggests that TabNet may not generalize well in low-data regimes.
>
> To test whether preprocessing choices could mitigate the observed ill-behavior, we evaluated three variants of TabNet: (1) no feature scaling with one-hot encoding, (2) no feature scaling with categorical input mode, and (3) standardized feature scaling with one-hot encoding. See the table below.
>
> | Ratio in CC18 (72 datasets)|Non-Monotone ($\neg$M)|Non-Convex ($\neg$C)|Ill-behaved ($\neg$M $\cup$ $\neg$C)|Peaking|Dipping|
> |-|-|-|-|-|-|
> |TabNet (noFS, onehot cate)|12.5%|65.3%|66.7%|19.4%|5.6%|
> |TabNet (noFS, input raw cate)|8.3%|69.4%|69.4%|27.8%|1.4%|
> |TabNet (standardFS, onehot cate)|16.7%|65.3%|68.1%|4.2%|2.8%|
>
> All variants show similarly high rates of ill-behavior. One could argue that early stopping might reduce overfitting and thus improve learning curve quality. However, we chose not to apply early stopping for consistency and fairness across learners, as it could potentially benefit all iterative models in LCDB 1.1.
>
> Concluding; we find that modern tabular models can also show more ill-behavior than expected. Modern deep learning methods Real MLP and ensemble methods as CatBoost show relatively well-behaved learning curves.
>
> ### 5. Statistical Correction
> **Did you consider or perform an analysis using a less stringent correction method, such as controlling for the False Discovery Rate (FDR), and if so, did it meaningfully change the 14% ill-behavior estimate?**
>
> We have performed additional analysis using slightly less conservative method called Holm’s Step-Down Procedure (Holm's method) [7], in the sense that it will reject more null hypotheses, typically resulting in fewer Type II errors. As shown in the tables below, the results remain consistent with our original findings: an even larger fraction of learning curves are  identified as ill-behaved, up to 19%. While a comprehensive exploration of alternative statistical corrections is beyond the scope of this paper, we recognize that this may be of interest to the community and will include these additional results in the appendix.
>
> | LCDB 1.1 CC18 (no feature scaling, 72 datasets) | Bonferroni Correction | Holm's Method |
> |-|-|-|
> |Non-Monotone ($\neg$M)|11.1%|13.6%|
> |Non-Convex ($\neg$C)|10.3%|14.7%|
> |Ill-behaved ($\neg$M $\cup$ $\neg$C)|14.3%|18.8%|
>
> | LCDB 1.1 FULL (no feature scaling, 265 datasets) | Bonferroni Correction | Holm's Method |
> |-|-|-|
> |Non-Monotone ($\neg$M)|10.4%|13.4% |
> |Non-Convex ($\neg$C)|10.8%|14.5%|
> |Ill-behaved ($\neg$M $\cup$ $\neg$C)|14.4%|19.0%|
>
> Our core message remains unchanged: even under the most conservative assumptions (Bonferroni), the proportion of ill-behaved learning curves is higher than previously believed.
>
> **References**
>
> [1] Kohli, R., Feurer, M., Eggensperger, K., Bischl, B., & Hutter, F. (2024). Towards quantifying the effect of datasets for benchmarking: A look at tabular machine learning. ICLR Workshop.
>
> [2] Prokhorenkova, L., Gusev, G., Vorobev, A., Dorogush, A. V., & Gulin, A. (2018). CatBoost: unbiased boosting with categorical features. NeurIPS.
>
> [3] Arik, S. Ö., & Pfister, T. (2021). Tabnet: Attentive interpretable tabular learning. AAAI.
>
> [4] Holzmüller, D., Grinsztajn, L., & Steinwart, I. (2024). Better by default: Strong pre-tuned mlps and boosted trees on tabular data. NeurIPS.
>
> [5] Hollmann, N., Müller, S., Purucker, L., Krishnakumar, A., Körfer, M., Hoo, S. B., ... & Hutter, F. (2025). Accurate predictions on small data with a tabular foundation model. Nature.
>
> [6] Erickson, N., Purucker, L., Tschalzev, A., Holzmüller, D., Desai, P. M., & Hutter, F. (2025). TabArena: A Living Benchmark for Machine Learning on Tabular Data. arXiv preprint arXiv:2506.16791.
>
> [7] James, G., Witten, D., Hastie, T., & Tibshirani, R. (2013). An introduction to statistical learning: with applications in R. Springer.

---

> > ### Comment · Reviewer_n6w6 · 2025-08-04
> > **Response for Rebuttal**
> >
> > Dear the authors,
> >
> > Thank you for your thorough and constructive rebuttal. I have reviewed your responses and am pleased to confirm that you have addressed all of my concerns, in many cases exceeding my expectations.
> >
> > I believe the following actions in particular significantly improve the value and credibility of your work:
> >
> > - Changing the license to the community-friendly CC BY 4.0.
> >
> > - Conducting an additional statistical analysis with Holm's method in response to the feedback, which further strengthens the paper's claims.
> >
> > - Expanding the scope of the paper by running additional experiments with modern learners like TabNet.
> >
> > - Committing to concrete improvements for the dataset's usability, such as adding a manifest file.
> >
> > I also want to add that your thoughtful response to the "Gaps in Responsible AI Documentation" was particularly thought-provoking. It served as a powerful reminder of the fundamental challenge of ensuring fairness and mitigating bias in datasets—a crucial issue in a scientific context where robust inferences depend on the quality of the underlying data.
> >
> > For these reasons, the concerns from my initial review have been completely resolved.
> > I recommend this paper for acceptance.
> >
> > Sincerely,
> >
> > --Reviewer n6w6

---

### Official Review · Reviewer_FeDP · 2025-07-03

**Rating:** 4
**Confidence:** 2

**Summary:**

This paper introduces the LCDB 1.1, which is a sample-wise learning curve database for tabular data. Compared to version 1.0, this dataset has higher resolution data, use feature scaling, try to tackle of the problem of data leakage, and use more datasets and learners. The authors conduct dataset analyses and found out that ample-wise leaning curves are more ill-behaved (under the authors' definition) than previously understood, shown to be non-monotonicity or non-convexity. They state that the most well-behaved are tree-based and ensemble learners, and learners like Sigmoid SVM, LDA, QDA implementations show high rates of ill-behavior. The authors state that the ill-behaved curves make it difficult to conduct parametric curve fitting. Thus, by providing the benchmark LCDB 1.1, it would help people better understand the behavior of learning curves and the help develop more robust learning methods.

**Additional Feedback:**

* How could the LCDB 1.1 dataset be useful in a modern machine learning training pipeline?

**Dataset Code Accessibility:**

Yes

**Dataset Code Comments:**

* The dataset and the code is readily accessible with clear documentation.

**Ethical Comments:**

I see no ethical concerns.

**Ethical Considerations:**

No, there are no or only very minor ethics concerns

**Final Justification:**

This is an interesting work with intriguing potential for further study in related fields.
* The authors successfully addressed most of my concerns. They have added experiments on more modern machine learning models, provided some explanations for the ill-behavior, and given examples where this ill-behavior can occur even with no tunable hyperparameters.
* However, more work is still needed to demonstrate the usefulness of this approach to the general machine learning training process.

The authors have clearly explained their ideas and run additional experiments to further support their suggestions, thus I raise my rating. However, due to the remaining issues, I cannot increase the rating further. Therefore, I am giving my final score of 4.

**Limitations Weaknesses:**

* This paper focuses on classical learners and does not address large-scale deep learning curves. Do the paper's implications generalize to the learning curves of modern machine learning methods? How can insights from these simpler cases help with more modern machine learning models?
* In this paper, the authors mention that lots of curves are ill-behaved, but do not give a discussion on the reasons. A brief theoretical analysis explaining the causes of this behavior would strengthen the paper, or possibly address the ill-behavior problem.
* The analysis was conducted using the default hyperparameters for the learners in Scikit-learn. Although the authors mention investigating whether this ill-behavior persists under hyperparameter tuning as a next step, they acknowledge this is computationally expensive, as the current work already required 600K CPU hours. This leaves it as a question whether these ill-behaviors can be "tuned away" through hyperparameter tuning or advanced preprocessors.

**Strengths Contributions:**

* The authors provide many empirical analysis of the ill-behaved learning curves and find out approximately 14% of the learning curves are ill-behaved. It identifies specific learners that causing more of these behaviors.

* The authors clearly explain the downstream consequences of the ill-behaved curves. They show how these curves would affect the parametric curve fitting and how they would undermine the effectiveness of the multi-fidelity model selection techniques like Successive Halving. These are important findings that would bring people attention.

* The paper is well-written and organized. The dataset and the code have clear documentation and are ready for use.

---

> ### Author Rebuttal · Authors · 2025-07-30
>
> We thank the reviewer for thoughtful feedback and we are encouraged that our work is seen as novel, well-written and impactful.
>
> We have added modern deep learning methods to show the generality of our findings. Additionally, we will add a small theoretical discussion about known causes of ill-behavior, but furthermore we want emphasize that explaining the causes of this behavior is non-trivial and remains a challenging open research question.
>
> **Do the paper's implications generalize to the learning curves of modern machine learning methods?**
>
> In the meantime, we have extended LCDB 1.1 by incorporating a broader set of modern tabular learners: the boosting model CatBoost [1] (2018), deep learning models such as TabNet [2] (2021) and RealMLP [3] (2024), and the foundation model TabPFN v2[4] (2025). According to a very recent study TabArena[5], CatBoost remains a strong SOTA model by default, while RealMLP achieves SOTA performance after tuning and ensembling. TabNet is widely used as a deep learning baseline, and TabPFN v2 is a well-performing foundation model for tabular data. We will update the database (not allowed during peer review). However, to answer your question, we have already computed the statistics regarding ill-behavior, see below.
>
> | Ratio in CC18 (72 datasets) with no feature scaling | Non-Monotone ($\neg$M) | Non-Convex ($\neg$C) | Ill-behaved ($\neg$M $\cup$ $\neg$C)|Peaking|Dipping|
> |-|-|-|-|-|-|
> |CatBoost|0.0%|0.0%|0.0%|0.0%|0.0%|
> |RealMLP|2.8%|1.4%|4.2%|0.0%|0.0%|
> |TabNet|12.5%|65.3%|66.7%|19.4%|5.6%|
> |TabPFN v2 (63 datasets)|0.0%|0.0%|0.0%|0.0%|0.0%|
>
> Note that, for TabPFN v2, it is only evaluated on a subset (63 datasets) of CC18 (72), because it only supports datasets with up to 10k training samples, 500 features, and 10 classes for classification task. For this rebuttal we only report results for CC18 with no feature scaling, but after the rebuttal phase we will generate all results for LCDB 1.1 full. Note that, TabPFN v2 and Real MLP used some of the datasets in CC18 during model development (e.g. tuning hyperparameters).
>
> In the results of our manuscript (see Table 3), we found that ensemble methods and tree-like learners have relatively little ill-behavior. Following this, we expected that CatBoost would show similar statistics as the Gradient Boosting learner of Scikit-learn, and indeed we see that it has very little ill-behavior as well. For the deep learner Real MLP, we find limited ill-behavior; we suspect that the substantial tuning of the hyperparameters that was part of Real MLP's design has alleviated a lot of the ill-behavior that remains present in our MLP baseline (which had 27% ill-behaving curves).
>
> We are not aware of any other works that generate learning curves for tabular foundation models, and as such, we were not very certain what to expect. TabPFN v2, according to the paper's title, was designed for small datasets, and is a Bayesian model which can alleviate overfitting. As such, it may be well-suited to learning curve analysis. This agrees with our findings of very little ill-behavior. In contrast, TabNet shows substantial ill-behavior, including various phase transitions. Clearly, this model was developed for larger datasets, because for larger sample sizes, often the learning curve improves significantly. It suggests that TabNet may not generalize well in low-data regimes.
>
> To test whether preprocessing choices could mitigate the observed ill-behavior, we evaluated three variants of TabNet: (1) no feature scaling with one-hot encoding, (2) no feature scaling with categorical input, and (3) standardized feature scaling with one-hot encoding. See the table below.
>
> | Ratio in CC18 (72 datasets)|Non-Monotone ($\neg$M)|Non-Convex ($\neg$C)|Ill-behaved ($\neg$M $\cup$ $\neg$C)|Peaking|Dipping|
> |-|-|-|-|-|-|
> |TabNet (noFS, onehot cate)|12.5%|65.3%|66.7%|19.4%|5.6%|
> |TabNet (noFS, input raw cate)|8.3%|69.4%|69.4%|27.8%|1.4%|
> |TabNet (standardFS, onehot cate)|16.7%|65.3%|68.1%|4.2%|2.8%|
>
> All variants show similarly high rates of ill-behavior. One could argue that early stopping might reduce overfitting and thus improve learning curve quality. However, we chose not to apply early stopping for consistency and fairness across learners, as it could potentially benefit all iterative models in LCDB 1.1.
>
> Concluding; we find that modern tabular models can also show more ill-behavior than expected. Modern deep learning methods Real MLP and ensemble methods as CatBoost show relatively well-behaved learning curves.
>
> **How could the LCDB 1.1 dataset be useful in a modern machine learning training pipeline? / ... help with more modern machine learning models?**
>
> Firstly, we'd like to underline the importance of studying simple and classical methods. These methods, that are known for decades, still seem to offer surprising learning curves. We believe that studying the learning curves from such simple learners, will also offer insights for learning curves of more complex and modern learning algorithms -- after all, if we cannot understand simple learners, how can we hope to understand complex ones?
>
> Additionally, these simple models are still relevant in modern pipelines, especially when little data is available (deep learning methods can struggle in these cases), or when interpretability and fairness is important, as is the case in high-stakes scenario's such as finance and healthcare (linear models such as logistic regression are ideal in such cases).
>
> The LCDB 1.1 can be integrated into modern machine learning pipelines in various ways. One of the most obvious ways, is for it to be used to develop modern multifidelity model selection algorithms and metalearning algorithms. Multifidelity algorithms (such as Successive Halving) use the training set size as a fidelity; and aim to speed up model selection by training models on smaller training sets. The LCDB 1.1 offers a very detailed dataset to develop and benchmark such multifidelity algorithms. Our Successive Halving experiment illustrates that learning curves in terms of the training set offer unique challenges due to learning curve crossings. As such, it will offer a fruitful dataset to develop new methods and benchmark existing ones.
>
> We also want to raise awareness: when developing new methods, one should not only focus on the performance for a single training set size. Instead, it is interesting to benchmark new learners for various dataset sizes, to understand its performances for various data regimes (i.e., learning curve)[6].
>
> **A brief theoretical analysis explaining the causes of this behavior would strengthen the paper, or possibly address the ill-behavior problem.**
>
> We agree this would strengthen the paper. However, it should be clear that this is a non-trivial ask. We would like to point out that, that such theoretical analysis often needs to be carried out per learner, and it is not clear how that should be carried out for all learners. Only alone for the peaking phenomema, various papers have studied this from different viewpoints, see for example [7-11]. This paper [12] gives an more detailed overview of all known theory papers regarding ill-behavior of learning curves. However, all these works are still mostly in the realm of learning theory, far from practice.
>
> By openly publishing the LCDB 1.1, and underscoring that ill-behavior is more prominent than previously thought, we want to bring these learning phemomema under the attention of the broader machine learning community. We believe this will lead to more researchers looking into the various ill-behaviors observed, which will hopefully lead to a better understanding of these ill-behaviors and how to avoid and mitigate them.
>
> **Can these ill-behaviors be "tuned away" through hyperparameter tuning or advanced preprocessors**
>
> While large-scale hyperparameter tuning and preprocessing is required to fully answer this question, we provide a simple illustrative example: the Nearest Centroid classifier, which has virtually no tunable hyperparameters, yet consistently exhibits ill-behaved learning curves. This suggests that some ill-behaviors may be inherent to the learner’s inductive bias and cannot be easily addressed through hyperparameter tuning.
>
> **References**
>
> [1] Prokhorenkova, L., Gusev, G., Vorobev, A., Dorogush, A. V., & Gulin, A. (2018). CatBoost: unbiased boosting with categorical features. NeurIPS.
>
> [2] Arik, S. Ö., & Pfister, T. (2021, May). Tabnet: Attentive interpretable tabular learning. AAAI.
>
> [3] Holzmüller, D., Grinsztajn, L., & Steinwart, I. (2024). Better by default: Strong pre-tuned mlps and boosted trees on tabular data. NeurIPS.
>
> [4] Hollmann, N., Müller, S., Purucker, L., Krishnakumar, A., Körfer, M., Hoo, S. B., ... & Hutter, F. (2025). Accurate predictions on small data with a tabular foundation model. Nature.
>
> [5] Erickson, N., Purucker, L., Tschalzev, A., Holzmüller, D., Desai, P. M., & Hutter, F. (2025). TabArena: A Living Benchmark for Machine Learning on Tabular Data. arXiv preprint arXiv:2506.16791.
>
> [6] Hoiem, D., Gupta, T., Li, Z., & Shlapentokh-Rothman, M. (2021, July). Learning curves for analysis of deep networks. ICML.
>
> [7] Duin, R. P. (2000, September). Classifiers in almost empty spaces. ICPR. IEEE.
>
> [8] Zollanvari, A., James, A. P., & Sameni, R. (2020). A theoretical analysis of the peaking phenomenon in classification. Journal of Classification.
>
> [9] Raudys, S., & Duin, R. P. (1998). Expected classification error of the Fisher linear classifier with pseudo-inverse covariance matrix. Pattern recognition letters.
>
> [10] Advani, M. S., Saxe, A. M., & Sompolinsky, H. (2020). High-dimensional dynamics of generalization error in neural networks. Neural Networks.
>
> [11] Hastie, T., Montanari, A., Rosset, S., & Tibshirani, R. J. (2022). Surprises in high-dimensional ridgeless least squares interpolation. Annals of statistics.
>
> [12] Viering, T., & Loog, M. (2022). The shape of learning curves: a review. IEEE TPAMI.

---

> > ### Comment · Reviewer_FeDP · 2025-08-03
> >
> > I thank the authors for the detailed response to my questions. It helped me to understand this work better. I will increase my score by 1.

---

### Decision · Program_Chairs · 2025-09-18

**Decision:**

Accept (poster)

**Comment:**

This paper presents new findings about the behavior of learning curves in machine learning. This work is based on the new database LCDB 1.1 providing high-resolution learning curves, and analyzes the factors that contribute to the ill-behavior of learners.


The reviewers found the work relevant and well-organized.  Major concerns were on the limitations of the study, including not investigating whether ill-behavior persists under hyperparameter optimization, and the lack of generalization of these findings to large-scale deep learning learning curves.


During the rebuttal, the authors acknowledged that a general theoretical analysis of ill-behaved learning curves is a non-trivial task and may require further work. However, while recognizing this limitation, they believe that publishing their findings on ill-behavior in learning curves can bring attention to this important area of research and potentially lead to future studies. The rebuttal also provides new experiments including modern learners, broadening the scope their work. They finally stress that this work can already contribute to a better understanding of the underlying mechanisms and potentially improve the performance of deep neural networks.


The reviewers appreciated the additional context and clarification of the rebuttal confirmed their favorable attitude towards the paper. As a result, the reviewers maintained or even increased their initial score, demonstrating a general agreement on the quality and merit of the paper. For this reason, the paper can be accepted to NeurIPS 2025 Datasets and Benchmarks Track.